

# Uncovering chemical signatures of salinity gradients through compositional analysis of protein sequences

Jeffrey M. Dick[1,2], Miao Yu[1], and Jingqiang Tan[1]

[1]Key Laboratory of Metallogenic Prediction of Nonferrous Metals and Geological Environment Monitoring, Ministry of Education, School of Geosciences and Info-Physics, Central South University, Changsha 410083, China
[2]State Key Laboratory of Organic Geochemistry, Guangzhou Institute of Geochemistry, Chinese Academy of Sciences, Guangzhou 510640, China

**Correspondence:** J. M. Dick (jeff@chnosz.net) or M. Yu (yumiao1987@pku.edu.cn)

**Abstract.** Thermodynamic influences on the chemical compositions of proteins in nature have remained enigmatic despite much work that demonstrates the impact of environmental conditions on amino acid frequencies. Here, we present evidence that the dehydrating effect of salinity is detectable as chemical differences in protein sequences inferred from 1) metagenomes and metatranscriptomes in regional salinity gradients and 2) differential gene and protein expression in microbial cells under hyperosmotic stress. The stoichiometric hydration state ($n_{H_2O}$), derived from the number of water molecules in theoretical reactions to form proteins from a particular set of basis species (glutamine, glutamic acid, cysteine, $O_2$, $H_2O$), decreases along salinity gradients including the Baltic Sea and Amazon River and ocean plume and in particle-associated compared to free-living fractions. However, the proposed metric does not behave as expected for hypersaline environments. Analysis of data compiled for hyperosmotic stress experiments under controlled laboratory conditions shows that differentially expressed proteins, as well as proteins coded by differentially expressed transcripts, are on average shifted toward lower $n_{H_2O}$. Notably, the dehydration effect is stronger for most organic solutes compared to NaCl. This new method of compositional analysis can be used to identify possible thermodynamic effects in the distribution of proteins along chemical gradients at a range of scales from biofilms to oceans.

## 1   Introduction

How microbial communities adapt to environmental gradients is a major challenge at the intersection of geochemistry, microbiology, and biochemistry. Patterns of amino acid usage in proteins are important indicators of microbial adaptation, and amino acid composition at the genome level is well known to depend on growth temperature (Zeldovich et al., 2007). Furthermore, measures of evolutionary distance and community composition based on protein sequences predicted from metagenomic sequencing are strongly associated with environmental temperature and pH (Alsop et al., 2014). It is widely acknowledged that the effect of amino acid substitutions on the structural stability of proteins is a major factor affecting amino acid usage in thermophiles (Sterner and Liebl, 2001; Zeldovich et al., 2007). Similarly, a large body of work has demonstrated amino acid signatures associated with proteins from halophilic organisms (Kunin et al., 2008; Paul et al., 2008; Oren, 2013; Boyd et al., 2014). The most common interpretation of these trends is that particular amino acid substitutions are selected through evolu-




tion to increase the stability and solubility of the folded conformation and enhance other structural properties such as flexibility
(Paul et al., 2008).

A complementary approach to interpreting patterns of amino acid composition is based on the energetics of amino acid synthesis. Energetic costs in terms of ATP requirements have been used to model protein expression levels in bacterial and yeast cells (Akashi and Gojobori, 2002; Wagner, 2005). Although ATP demands depend on environmental conditions (Akashi and Gojobori, 2002), a limitation of ATP-based models is that they are derived for specific biosynthetic pathways, such as whether
cells are grown in respiratory or fermentative (i.e. aerobic or anaerobic) conditions (Wagner, 2005). A different class of models, based on thermodynamic analysis of the overall Gibbs energy of reactions to synthesize metabolites from inorganic precursors, quantifies the energetics of the reactions in terms of temperature, pressure, and chemical activities of all the species in the reactions, including those that define pH and oxidation-reduction potential (Shock et al., 2010). Notably, the overall energetics for amino acid synthesis become more favorable, but to a different extent for each amino acid, between cold, oxidizing seawater
and hot, reducing hydrothermal solution (Amend and Shock, 1998). A recent systems biology study demonstrates tradeoffs between Gibbs energy of alternative pathways for amino acid synthesis and cofactor use efficiency (which affects ATP costs) in the model organism *Escherichia coli* and suggests that pathway thermodynamics play a role in thermophilic adaptation (Du et al., 2018). Nevertheless, energetic models have not made much headway in relating metagenomic and geochemical data. This may be because few studies have asked whether specific changes in the chemical composition of biomolecules reflect
specific environmental conditions.

To help close this gap, here we use compositional analysis of protein sequences to identify chemical signatures of two types of environmental conditions: redox and salinity gradients. Because redox reactions are inherent in many aspects of metabolism, while hydration and dehydration reactions are essential for the synthesis of biomacromolecules (Braakman and Smith, 2013), our approach is shaped by the assumption that $O_2$ and $H_2O$ are two primary components that link environmental conditions
to the energetics of biomolecular synthesis. Thermodynamic considerations predict that redox gradients supply a driving force for changes in the oxidation state of biomolecules (similar reasoning applies to the oxygen content of proteins; Acquisti et al., 2007), while salinity gradients, through the dehydrating potential associated with osmotic effects, exert a force that selectively alters the hydration state of biomolecules.

To test these predictions, we used two compositional metrics, the carbon oxidation state ($Z_C$) and stoichiometric hydration
state ($n_{H_2O}$). $Z_C$ is computed from the chemical formulas of organic molecules, and takes values between the extremes of -4 for $CH_4$ and +4 for $CO_2$, although the range for particular classes of biomolecules is much smaller (Amend et al., 2013). $n_{H_2O}$ is derived from the number of water molecules in theoretical formation reactions of proteins from basis species (Dick, 2016, 2017). Through the compositional analysis of representative metagenomic and metatranscriptomic datasets, we show that $Z_C$ and $n_{H_2O}$ are most closely aligned with environmental redox and salinity gradients, respectively. These findings apply
to freshwater and marine environments, but trends for hypersaline environments deviate from the thermodynamic predictions, most likely due to evolutionary optimizations of hydrophobicity and isoelectric point to stabilize the structures of proteins in halophilic organisms.



In a previous study (Dick et al., 2019), we compared one broad class of geochemical conditions (redox gradients) with one compositional metric for proteins (carbon oxidation state). Here, we expand the geobiochemical framework to two dimensions by considering another set of environments (salinity gradients) and another compositional metric (stoichiometric hydration state). A long-term research goal is to extend this framework to as many dimensions as there are thermodynamic components plus temperature and pressure.

## 2 Conceptual background

In this study we use compositional analysis to uncover environmental imprints in protein sequences. Analysis of compositional data is used by geochemists to study processes such as water-rock interaction and ore deposition, and is often one of the first steps in constructing thermodynamic models, but its application to living systems is relatively uncommon. Therefore, it is important to describe the conceptual basis for our methods. To do this, we identified six areas of concern posed as alternatives: 1) intracellular or environmental conditions, 2) amino acids or atoms, 3) condensation or theoretical formation reactions, 4) chemical composition or conformational stability, 5) oxidation and hydration state or temperature and pH, and 6) mathematical or biosynthetic models.

A first concern is that intracellular conditions are maintained within physiological ranges, so the influence of external conditions on the composition of microbial biomolecules may be limited. However, cell membranes are permeable to uncharged species such as hydrogen (Slonczewski et al., 2009), supporting the argument that the oxidation state of the cytoplasm, and therefore the energetics of metabolic reactions, are influenced by the external environment (Poudel et al., 2018; Canovas and Shock, 2020). Likewise, oxygen diffuses rapidly through lipid membranes, depending on their composition and structure, and rates of diffusion increase with temperature (Möller et al., 2016). Cell membranes are also permeable to water (Record et al., 1998). For *E. coli*, which grows most rapidly at about 0.3 OsM (osmolarity), increasing the extracellular osmotic strength from 0.1 to 1.0 OsM [approximately the osmotic concentration of seawater; BioNumbers BNID 100802 (Milo et al., 2010)] reduces the amount of free cytoplasmic water by more than half (Record et al., 1998). Halophiles, which thrive at even higher salinities, accumulate inorganic salts or organic solutes to maintain osmotic balance with the environment (Garner and Burg, 1994; Oren, 2013). The result is that, with few exceptions, intracellular conditions must be isosmotic with the environment, or somewhat higher to maintain turgor pressure (Gunde-Cimerman et al., 2018). Water activity is lower in more concentrated solutions, and intracellular water activity estimated from freezing point and cell composition data closely follows that of the growth medium, but is often offset to lower values (Chirife et al., 1981), perhaps due to macromolecular crowding effects (Garner and Burg, 1994). In other words, high osmotic strength causes a decrease in hydration potential, measured as water activity, both outside and inside cells.

This brief review suggests that oxidation and hydration potentials in cell interiors, at least under experimental conditions, are influenced by, but not equal to, environmental conditions. Ideally, we would like to compare the compositions of biomolecules to conditions actually measured inside cells or in the immediate surroundings of cells, but these measurements are generally not available for microbial communities in their natural environments, so we make comparisons with large-scale geochemical





gradients, except for different layers of the Guerrero Negro microbial mat, where metagenomic and chemical data are available on the scale of millimeters.

Second, previous authors have emphasized the importance of changes in elemental stoichiometry – that is, atomic composition – and not only amino acid composition in the molecular evolution of proteins (Baudouin-Cornu et al., 2001). Although
stoichiometric predictions are amenable to experimental tests, such as the long-term evolution of *Escherichia coli* in the laboratory (Turner et al., 2017), the omission of a major bioelement, hydrogen, and the oxidation state of organic matter from most stoichiometric models (Karl and Grabowski, 2017) means that there are also significant opportunities for theory development.

The third point follows from the previous one. The polymerization of amino acids is a condensation reaction that releases one $H_2O$ per bond formed, independent of the particular amino acids that are involved. By contrast, our analysis depends
crucially on the concept of a "formation reaction", which in the thermodynamic literature represents the composition of a chemical species, either in terms of elements (Warn and Peters, 1996), or in terms of other species (May and Rowland, 2018). When these other species are restricted in number to the minimum needed to represent the composition of all possible species in the system, they constitute a set of "basis species", which can be thought of as the building blocks of the system, similar to the concept of thermodynamic components (Anderson, 2005). Therefore, a formation reaction from basis species is a mass-
balanced, but non-unique, stoichiometric representation of the chemical composition of the protein. This type of reaction in general does not correspond to amino acid biosynthesis or polymerization, so to avoid confusion, we refer to these formation reactions as "theoretical formation reactions"; the number of water molecules in the theoretical formation reactions is the "stoichiometric hydration state".

From a mechanistic standpoint, an analysis using any set of basis species is inadequate, since the number of basis species
(five, corresponding to the elements C, H, N, O, and S) is smaller than the number of biochemical precursors and inorganic species that are actually involved in amino acid synthesis (Du et al., 2018). The use of $O_2$, $H_2O$, and other basis species to represent the composition of proteins reflects the hypothesis that they are conjugate to thermodynamically meaningful descriptive variables (specifically, chemical potentials) even if they are not directly involved in the biosynthetic mechanisms for amino acids. The projection of amino acid composition (20-D) into the compositional space represented by basis species (5-
D) is a type of dimensionality reduction, but the variables are chosen based on a physicochemical hypothesis, unlike principal components analysis (PCA) or other unsupervised methods, where the projection is determined by the data.

A fourth concern is that this analysis is based on the hypothesis that thermodynamic forces affect the chemical compositions of proteins over evolutionary time, which is different from the more common hypothesis of optimization of structural stability. Thermodynamic models define the "cost" of a protein as a function of not only amino acid composition but also environmental
conditions. Conceptually, this follows from Le Chatelier's principle, in that increasing the chemical activity of a reactant (on the left-hand side of a reaction) drives the reaction toward the products, or in more general terms, that the overall Gibbs energy of a reaction depends on the activities of species in the reaction (Shock et al., 2010; Amend and LaRowe, 2019). Consider two proteins with different amino acid compositions, and therefore also different chemical compositions and theoretical formation reactions, which should be normalized by the number of residues in order to compare proteins of different length. The formation
of the protein with more water as a reactant is theoretically favored by increasing the water activity, whereas the formation





of the protein with more oxygen as a reactant is favored by increasing the oxygen activity. The water and oxygen activity are thermodynamic measures of hydration and oxidation potential and can be converted to other scales, such as oxidation-reduction potential (ORP).

This reasoning provides the theoretical justification for using chemical composition as an indicator of molecular adaptation to specific environmental conditions, but does not replace interpretations based on structural considerations. Halophilic organisms exhibit well-documented patterns of amino acid usage, including lower hydrophobicity and higher abundance of acidic residues, that impart greater stability, solubility, and flexibility of proteins (Paul et al., 2008). These adaptations are reflected in lower values of the GRAVY hydrophobicity scale (Paul et al., 2008; Boyd et al., 2014) and/or isoelectric point of proteins (pI) (Oren, 2013). In Sect. 4.3 and 4.4, we compare the compositional metrics with GRAVY and pI for the same datasets.

Fifth, temperature, pH, and other environmental parameters besides redox and salinity might influence the oxidation and hydration state of proteins. For instance, the redox gradients in hydrothermal systems are also temperature gradients, due to the mixing of seawater and hydrothermal fluid, and we have not attempted to disentangle the effects of temperature and redox conditions. However, our previous analysis of other redox gradients, including stratified hypersaline lakes, indicates that carbon oxidation state of biomolecules can vary even in systems where temperature changes are much smaller (Dick et al., 2019). It is an axiomatic statement that changes in oxidation state can be associated with one thermodynamic component of a system; our objective in the present study is to explore the differences between this and one other component, represented by hydration state. Future work should also account for the effects of pH and temperature, which is possible using thermodynamic models for proteins (Dick and Shock, 2011).

Finally, it should be noted that the basis species used in the stoichiometric analysis are chosen primarily for mathematical convenience, not because of evolutionary or biosynthetic requirements. The basis species we use for deriving the stoichiometric hydration state of proteins are glutamine, glutamic acid, cysteine, $O_2$, and $H_2O$ (designated "QEC"). The primary reason for choosing these basis species is to reduce the covariation between the metrics for oxidation and hydration state; that covariation is a mathematical consequence of projecting the atomic formulas of proteins into a particular compositional space, and may not reflect meaningful differences of chemical composition. There is nothing implied by the choice of basis species about evolutionary or biosynthetic mechanisms, and any set of basis species is thermodynamically valid, as long as they are the minimum number needed to represent the chemical composition of all the species in the system (Anderson, 2005). However, it is most convenient to select basis species that correspond to the controlling variables of the system. The QEC basis species has a biological rationale since glutamine and glutamic acid are often identified as highly abundant metabolites and have been characterized as "nodal point" metabolites (Walsh et al., 2018). Other considerations are described in Sect. 3.2.

# 3 Methods

## 3.1 Carbon oxidation state

The most common metric used in geochemistry for the oxidation state of organic molecules is the average oxidation state of carbon ($Z_C$), which also goes by other names such as nominal oxidation state of carbon (NOSC) (LaRowe and Van Cappellen,





2011). This quantity measures the average degree of oxidation of carbon atoms in organic molecules. For a protein for which
the primary sequence has the chemical formula $C_cH_hN_nO_oS_s$, the value of $Z_C$ can be calculated from (Dick and Shock, 2011;
Dick, 2014)

$$Z_C = \frac{-h + 3n + 2o + 2s}{c} \tag{1}$$

The derivation of Eq. (1) is based on the relative electronegativities of the elements, expressed as oxidation numbers (e.g.
Kauffman, 1986; Minkiewicz et al., 2018). When bonded to carbon, H is assigned an oxidation number of +1, and N, O, and S
have oxidation numbers of -3, -2, and -2. Eq. (1) gives the remaining charge that must be present on each C atom, on average,
to satisfy overall neutrality. Because of the relatively simple structures of amino acids and the primary structure of proteins,
in which N, O, and S are bonded to only H and C, it is possible to calculate the average oxidation state of carbon using Eq.
(1). However, this equation is not necessarily valid for other classes of organic molecules or some types of post-translational
modifications of proteins, including the formation of disulfide bonds. An important relation given by Eq. (1) is the redox
neutrality of hydration and dehydration reactions; any pair of hypothetical (or real) proteins whose formulas differ only by
some amount of $H_2O$ have identical carbon oxidation states.

### 3.2 Choice of basis species

A major premise of this study is that oxidation state and hydration state are two primary variables in geobiochemical systems.
Accordingly, when choosing the basis species that can be combined to make the proteins, $O_2$ and $H_2O$ are the only fixed
requirements. This leaves three basis species that when combined with each other and with $O_2$ and $H_2O$ must be able to give
any possible formula written as $C_cH_hN_nO_oS_s$. Note again that this analysis refers to the chemical formulas of polypeptide
sequences, that is, the primary structure of proteins, not post-translational modifications or $H_2O$ molecules in the hydration
shell of folded proteins.

Eq. (1) is derived from electronegativity relations and therefore allows the calculation of the carbon oxidation state from
a given chemical formula, independent of any chemical reactions. In contrast, there is no way to count the number of $H_2O$
molecules in a chemical formula; $H_2O$ appears only in chemical reactions. But it is important to note that any particular
reaction that involves only $H_2O$ is redox-neutral. Extrapolation of this principle to the general case gives the criterion that a
metric for hydration state should be disconnected from redox effects. In other words, when applied to a population of target
molecules, such as all the proteins in a genome, the correlation between metrics for oxidation state and hydration state should
be minimized.

Accordingly, we aim to find a projection of the elemental composition of primary protein sequences that clearly separates
$Z_C$ and the stoichiometric number of $H_2O$. There are no thermodynamic restrictions on the choice of basis species, but a
biologically meaningful set is likely to comprise metabolites that have high network connectivity, that is, are involved in
reactions with many other metabolites. Reactions involving glutamine and glutamic acid, or its ionized form, glutamate, are
major steps of nitrogen metabolism (Morowitz, 1999; DeBerardinis and Cheng, 2010). Either methionine or cysteine would
provide the sulfur required for the system, but cysteine is relevant as a constituent of the glutathione molecule, which has





**Table 1.** Values of stoichiometric hydration state ($n_{H_2O}$) of amino acid residues calculated with the rQEC derivation. Standard one-letter abbreviations for the amino acids (AA) are used.

| AA | $n_{H_2O}$ | AA | $n_{H_2O}$ | AA | $n_{H_2O}$ | AA | $n_{H_2O}$ |
|----|-----|----|-----|----|-----|----|-----|
| A | 0.369 | G | 0.478 | M | 0.046 | S | 0.575 |
| C | -0.025 | H | -1.825 | N | -0.122 | T | 0.569 |
| D | -0.122 | I | 0.660 | P | -0.354 | V | 0.522 |
| E | -0.107 | K | 0.763 | Q | -0.107 | W | -4.087 |
| F | -2.568 | L | 0.660 | R | 0.072 | Y | -2.499 |

important roles in cellular redox chemistry (Walsh et al., 2018). These considerations support the proposal of the amino acids glutamine, glutamic acid, and cysteine (collectively abbreviated QEC) together with $O_2$ and $H_2O$ as a biologically relevant set of basis species for describing the chemical compositions of proteins (Dick, 2016). These three amino acids are among the top eight amino acids ranked by number of reactions in a metabolic model for *Escherichia coli* (Feist et al., 2007) (Glu: 52, Ser: 25, Asp: 23, Gln: 18, Ala: 15, Gly: 15, Met: 15, Cys: 13).

### 3.3 Derivation of stoichiometric hydration state

Here we compute the stoichiometric hydration state by analyzing the compositions of the 20 proteinogenic amino acids in detail. Using the basis species $CO_2$, $NH_3$, $H_2S$, $H_2O$, and $O_2$ (designated CHNOS), the theoretical formation reaction of alanine ($C_3H_7NO_2$) is

$$3CO_2 + 2H_2O + NH_3 \rightarrow C_3H_7NO_2 + 3O_2 \tag{R1}$$

and the oxygen and water content of the amino acid (i.e, $n_{O_2} = -3$ and $n_{H_2O} = 2$) are the opposite of the coefficients on $O_2$ and $H_2O$ in the reaction. Similar reactions for the other amino acids were used to make Fig. 1a–b. Using glutamine ($C_5H_{10}N_2O_3$), glutamic acid ($C_5H_9NO_4$), cysteine ($C_3H_7NO_2S$), $H_2O$, and $O_2$ (the QEC basis species), the theoretical formation reaction of alanine is

$$0.4C_5H_{10}N_2O_3 + 0.2C_5H_9NO_4 + 0.6H_2O \rightarrow C_3H_7NO_2 + 0.3O_2 \tag{R2}$$

showing that the oxygen and water content are $n_{O_2} = -0.3$ and $n_{H_2O} = 0.6$. Calculations for all the amino acids using the QEC basis were used to make Fig. 1c–f.

The CHNOS basis yields a strong negative correlation between $Z_C$ and $n_{H_2O}$ for the amino acids (Fig. 1a), but a relatively weak correlation between $Z_C$ and $n_{O_2}$ (Fig. 1b). The QEC basis provides a much stronger association between $Z_C$ and $n_{O_2}$ and greatly reduces the correlation between $Z_C$ and $n_{H_2O}$ (Fig. 1c–d). However, there is still a small negative correlation for amino acids (Fig. 1d), which is also visible in whole-proteome data for humans and *E. coli* (Fig. 1e–f). We calculated residual-corrected values of $n_{H_2O}$ by taking the residuals of a linear model for amino acids (Fig. 1d), then subtracting a constant, defined





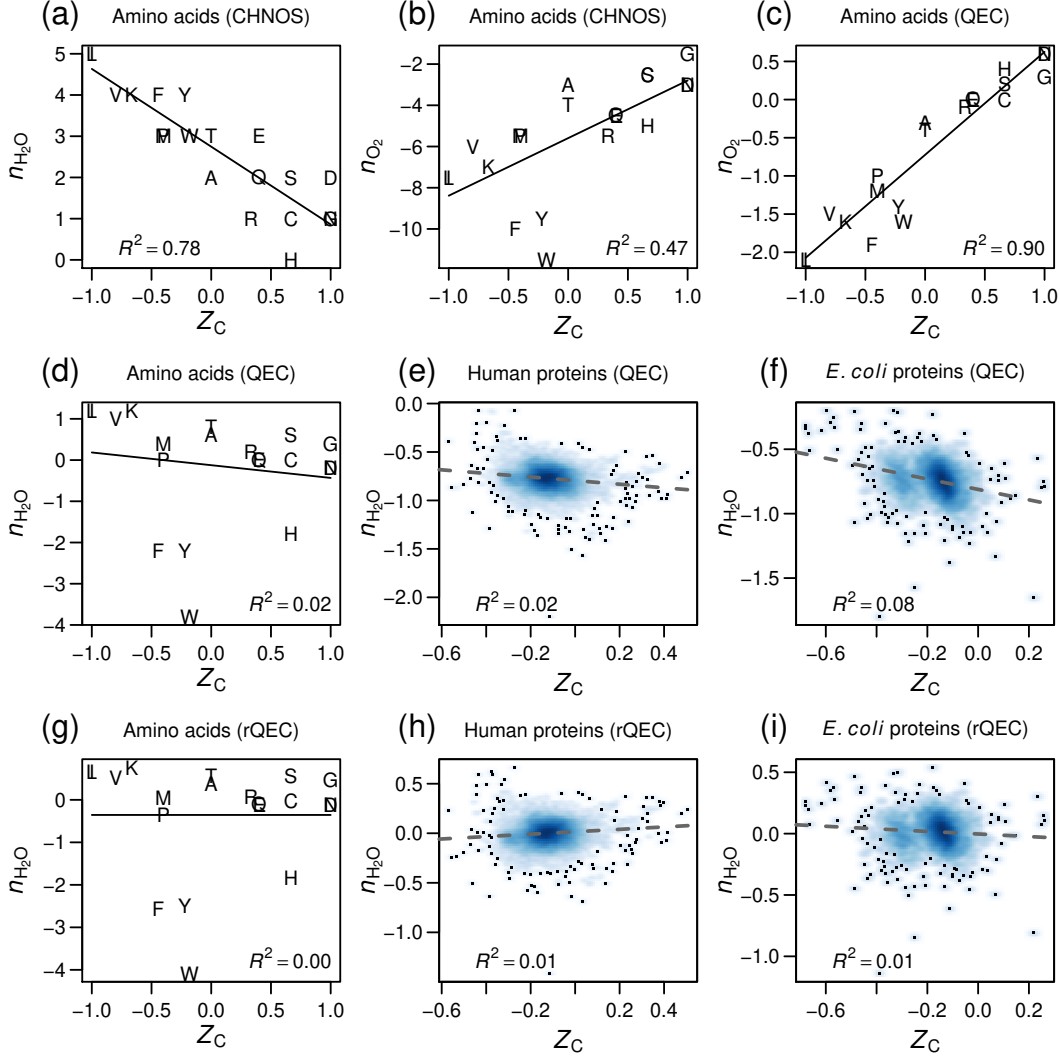

**Figure 1.** Stoichiometric values for theoretical formation reactions of amino acids computed with different sets of basis species (CHNOS and QEC) and derivation of the residual correction (rQEC). **(a–b)** Number of $H_2O$ and $O_2$ in the theoretical formation reactions of amino acids from $CO_2$–$NH_3$–$H_2S$–$H_2O$–$O_2$ (CHNOS) are plotted against carbon oxidation state ($Z_C$), which is also computed from the chemical formula but does not depend on the choice of basis species. Linear models and $R^2$ values were calculated using the `lm` function in R (R Core Team, 2020). **(c–d)** Changing the basis species to glutamine–glutamic acid–cysteine–$H_2O$–$O_2$ (QEC) strengthens the association between $Z_C$ and $n_{O_2}$ and decreases that between $Z_C$ and $n_{H_2O}$. However, there is still a noticeable negative correlation between $Z_C$ and $n_{H_2O}$, which is also visible in scatterplots of all proteins in **(e)** *H. sapiens* and **(f)** *E. coli* K12 [UniProt reference proteomes UP000005640 and UP000000625 (The UniProt Consortium, 2019)]. **(g)** Residuals from the linear model in **(d)** minus a constant of 0.355 yield values for the stoichiometric hydration state (rQEC) of amino acids. **(h–i)** Stoichiometric hydration states of proteins calculated with the rQEC values. The constant was defined so that the mean $n_{H_2O}$ for human proteins equals zero.





such that the mean $n_{H_2O}$ for all human proteins equals zero. This derivation, which we refer to as "rQEC", gives the residual-

corrected stoichiometric hydration state for each amino acid, which is plotted in Fig. 1g and listed in Table 1. Even with the residual correction for amino acids, there remain slightly positive and negative correlations for human and *E. coli* proteins (Fig. 1h–i). As noted above, the mean $n_{H_2O}$ for human proteins was defined to be zero; the mean for proteins in *E. coli* is somewhat greater, at 0.014.

By strengthening the association between $Z_C$ and $n_{O_2}$, which can both be interpreted as metrics for oxidation state, and

reducing the correlation between $Z_C$ and $n_{H_2O}$, the QEC basis species provides a more convenient projection of chemical composition than a "default" choice of inorganic species, such as $CO_2$, $NH_3$, $H_2S$, $H_2O$, and $O_2$, which commonly appear in overall catabolic reactions (Amend and LaRowe, 2019). Furthermore, the residual correction allows the identification of horizontal or vertical trends on $n_{H_2O}$–$Z_C$ scatterplots to be associated with changes in only oxidation state or hydration state, respectively.

**3.4 Compositional metrics for proteins and metagenomes**

For a given protein, the stoichiometric hydration state was calculated by taking the sum of (number of each amino acid multiplied by the respective value of $n_{H_2O}$ in Table 1), then dividing the result by the number of amino acids. The average oxidation state of carbon was also calculated from the values for the amino acids [see Table 1 of Dick and Shock (2011)]. Unlike $n_{H_2O}$, averages for $Z_C$ must be weighted by the number of carbon atoms in each amino acid. For example, $Z_C$ of the dipeptide Ala-Gly

can be calculated as $(3 \times 0 + 2 \times 1) / (3 + 2)$, where 3 and 2 are the numbers of carbon atoms and 0 and 1 are the $Z_C$ of Ala and Gly, respectively. The result, 0.4, can be checked by applying Eq. 1 to the chemical formula of alanylglycine ($C_5H_{10}N_2O_3$).

**3.5 Amino acid composition of proteomes of Nif-bearing organisms**

Amino acid compositions of all proteins for each bacterial, archaeal, and viral taxon in the NCBI Reference Sequence (RefSeq) database (O'Leary et al., 2016) were compiled from RefSeq release 95 (July 2019). Scripts to do this, and the resulting data

file of amino acid compositions of 36,425 taxa, are available in the JMDplots R package (see *Code and data availability*). Names of organisms containing different nitrogenase (Nif) homologs were extracted from Supplemental Table 1A of Poudel et al. (2018). These names were matched to the closest organism name in RefSeq. Duplicated species (represented by different strains) were removed, as were matching organisms with fewer than 1000 RefSeq protein sequences. As a result, the numbers of organisms included in the present calculations (Nif-A: 157, Nif-B: 69, Nif-C: 14, Nif-D: 7) are less than those identified

in Poudel et al. (2018). Note that values of $Z_C$ calculated here (Fig. 2a) are lower than those shown in Fig. 5 of Poudel et al. (2018). This difference is associated with the weighting by carbon number (described above), which was not performed by Poudel et al. (2018).





### 3.6  GRAVY and pI

The grand average of hydropathicity (GRAVY) was calculated using published hydropathy values for amino acids (Kyte and
Doolittle, 1982). The isoelectric point was calculated using published p$K$ values for terminal groups (Bjellqvist et al., 1993)
and sidechains (Bjellqvist et al., 1994); however, the calculation does not implement position-specific adjustments (Bjellqvist
et al., 1994). The charge for each ionizable group was precalculated from pH 0 to 14 at intervals of 0.01, and the isoelectric
point was computed as the pH where the sum of charges of all groups in the protein is closest to zero. These calculations
were implemented as new functions in the canprot R package (Dick, 2017) (see *Code and data availability*). Comparisons
for selected proteins show that the calculated values of GRAVY and pI are equal to those obtained with the ProtParam tool
(Gasteiger et al., 2005).

### 3.7  Prediction of protein sequences

Protein sequences were predicted from metagenomic reads using a previously described workflow (Dick et al., 2019). Briefly,
reads were trimmed, filtered, and dereplicated using scripts adapted from the MG-RAST pipeline (Keegan et al., 2016). For
metatranscriptomic datasets, ribosomal RNA sequences were removed using SortMeRNA (Kopylova et al., 2012). Protein-
coding sequences were identified using FragGeneScan (Rho et al., 2010), and the amino acid sequences of the predicted
proteins were used in further calculations. For large datasets, only a portion of the available reads was processed (at least
500,000 reads; see Supplementary Tables S1 and S2). This reduces the computational requirements without noticeably affecting
the calculated average compositions (Dick et al., 2019).
Means and standard deviations of $Z_C$, $n_{H_2O}$, GRAVY, and pI were calculated for 100 random subsamples of protein sequences
from each metagenomic or metatranscriptomic dataset. The numbers of sequences included in the subsamples were chosen to
give a total length closest to 50,000 amino acids on average.

## 4  Results and discussion

### 4.1  Comparison of redox and salinity gradients

To search for the hypothesized dehydration signal in metagenomic data, we began with redox gradients as a negative control.
Submarine hydrothermal vents are zones of complex interactions between reduced endmember fluids and relatively oxidized
seawater (Reeves et al., 2014; Ooka et al., 2019). Terrestrial hydrothermal systems, such as the hot springs in Yellowstone
National Park, USA, provide a source of reduced fluids that are oxidized by degassing and mixing with air and surface ground-
water as well as biological activity including sulfide oxidation (Lindsay et al., 2018). Redox gradients can also develop over
smaller length scales. The surface of the Guerrero Negro microbial mat (Baja California Sur, Mexico) is exposed to ca. 1
m deep hypersaline, oxygenated water (approximately 200 μM O$_2$), but in the mat, oxygen rises during the daytime and is
depleted within a few millimeters, giving way to anoxic, then sulfidic conditions (Ley et al., 2006).

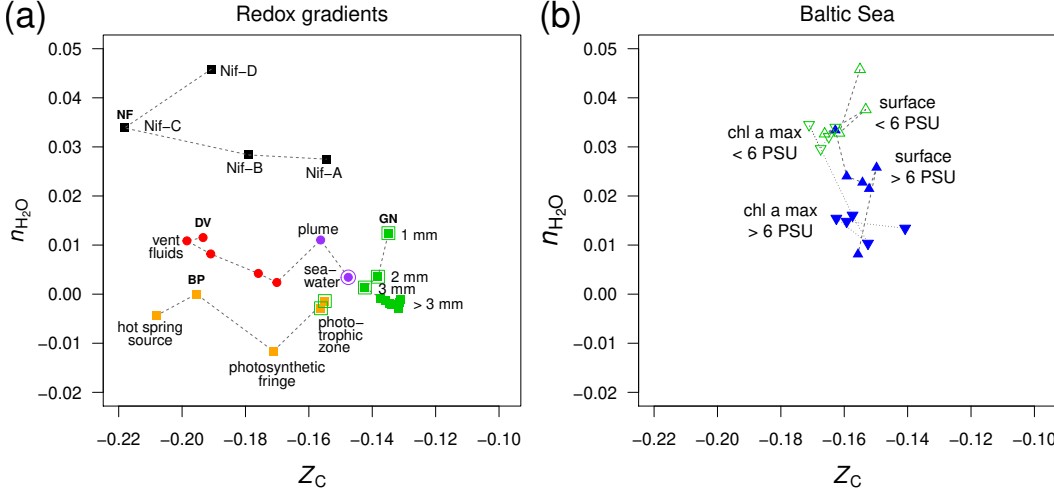

**Figure 2.** Compositional analysis of proteins in redox gradients and the Baltic Sea salinity gradient. **(a)** Redox gradients. Abbreviations and data sources are given in Fig. 2. Outlined symbols indicate samples in relatively oxidizing conditions. **(b)** Surface and deeper samples (chl a max: chlorophyll a maximum, 9–30 m deep) from the Baltic Sea transect. Metagenomes as described in Dupont et al. (2014) were downloaded from iMicrobe (Youens-Clark et al., 2019); data for the 0.1–0.8 μm size fraction are plotted here. Upward- and downward-pointing symbols, connected by dashed and dotted lines, represent surface and deeper samples, respectively, from stations along the transect at low salinity (< 6 PSU) and high salinity (> 6 PSU).

Using metagenomic data for these redox gradients (Kunin et al., 2008; Havig et al., 2011; Swingley et al., 2012; Reveillaud et al., 2016; Fortunato et al., 2018), Dick et al. (2019) showed that the carbon oxidation states of DNA, messenger RNA, and

proteins increase down the outflow channel of Bison Pool and between fluids from diffuse hydrothermal vents and relatively oxidizing seawater. Notably, intact polar lipids extracted from the microbial communities of Bison Pool and other alkaline hot springs also exhibit downstream increases in carbon oxidation state (Boyer et al., 2020), confirming that similar trends characterize multiple classes of biomolecules. The $Z_C$ of proteins increases more subtly toward the surface in the upper few millimeters of the Guerrero Negro microbial mat; it also increases at greater depths, perhaps due to heterotrophic degradation

and/or horizontal gene transfer (Dick et al., 2019). Furthermore, an evolutionary trajectory associated with the occurrence of different homologs of nitrogenase (Nif) in anaerobic and aerobic organisms is characterized by increasing $Z_C$ of the proteomes of these organisms (Poudel et al., 2018).

The trends described above are visible in the $n_{H_2O}$–$Z_C$ scatter plot in Fig. 2a. With the exception of Guerrero Negro, these datasets exhibit larger changes in carbon oxidation state than stoichiometric hydration state. This is an expected outcome, as

the redox gradients considered here do not have large changes in salinity. In particular, concentrations of Cl⁻, a conservative ion, increase by less than 10% (6.1 to 6.6 mM) in the outflow of Bison Pool due to evaporation (Swingley et al., 2012). The diffuse vents considered here have concentrations of Cl⁻ between 515 and 624 mM, not greatly different from bottom seawater at 545 mM [Dataset S1 of Reeves et al. (2014)].

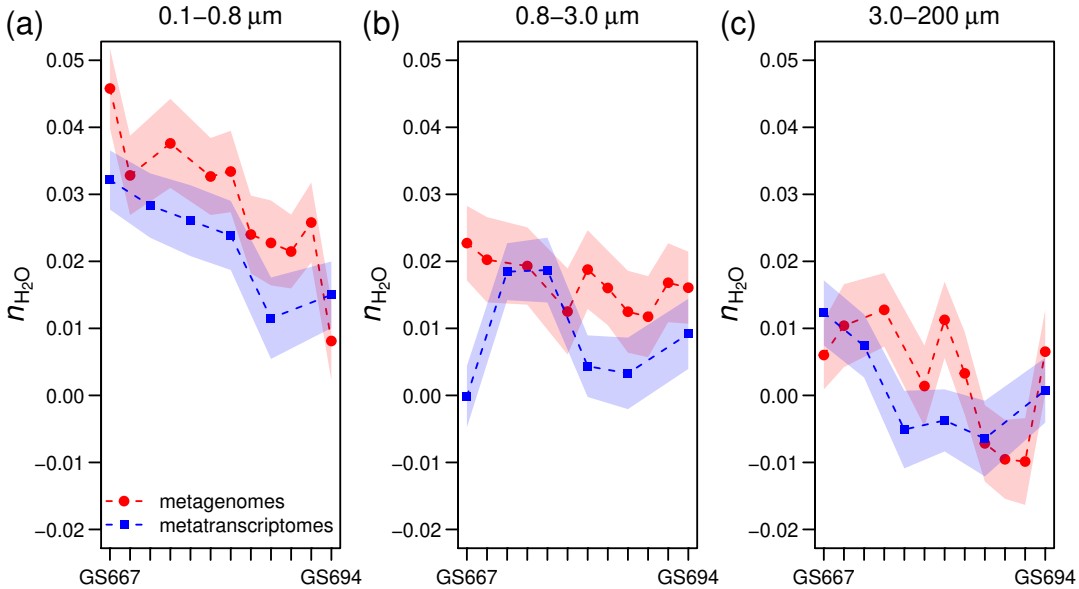

**Figure 3.** Stoichiometric hydration state of proteins in metagenomes (Dupont et al., 2014) and metatranscriptomes (Asplund-Samuelsson et al., 2016) of surface water samples in the Baltic Sea with increasing particle size: **(a)** 0.1–0.8 µm, **(b)** 0.8–3.0 µm, **(c)** 3.0–200 µm. From left to right, the samples on the *x*-axis (some IDs omitted for clarity) are arranged from freshwater to marine conditions in the Sorcerer II Global Ocean Sampling Expedition (Dupont et al., 2014); all sample IDs are GS667, GS665, GS669, GS673, GS675, GS659, GS679, GS681, GS683, GS685, GS687, GS694. Width of shading represents ±1 standard deviation in subsampled sequences (see Methods).

As a well-known example of a regional salinity gradient, the Baltic Sea exhibits a freshwater to marine transition over 1800

km, but dissolved oxygen at the surface is at or near saturation with air (Dupont et al., 2014), so this transect does not represent a redox gradient. For protein sequences derived from metagenomes in the 0.1–0.8 µm size fraction, there are large changes in stoichiometric hydration state along the Baltic Sea transect, but relatively small differences in the carbon oxidation state (Fig. 2b). This pattern holds for samples from both the surface and chlorophyll a maximum (9–30 m deep).

### 4.2   Multifactorial hydration effects

Metagenomic and metatranscriptomic data for different filter size fractions are available for the Baltic Sea. The 0.1–0.8 µm and 0.8–3.0 µm size fractions represent free living bacteria, while the 3.0–200 µm fraction contains particle-associated bacteria with average larger genome sizes and greater inferred metabolic and regulatory capacity (Dupont et al., 2014). Figure 3 shows that proteins inferred from metagenomes for larger particles have lower $n_{H_2O}$ than those for the smallest size fraction. The Guerrero Negro microbial mat offers another opportunity to compare exposed and interior environments. Unlike $Z_C$, which

reaches a minimum a few millimeters into the mat, $n_{H_2O}$ decreases throughout the mat, but the changes are most pronounced in the upper few millimeters (Fig. 2a).





One hypothesis that could explain these findings is that the interiors of particles and the mat are sequestered to some extent from the surrounding aqueous environment. If limited accessibility to the aqueous phase were manifested as lower water activity [perhaps due to surface effects associated with geological nanomaterials (Wang et al., 2003) and/or higher concentra-
tions of solutes], it would provide a thermodynamic drive that favors lower $n_{H_2O}$ of proteins. However, it should be noted that particles are also suitable habitats for multicellular and eukaryotic populations (Simon et al., 2014). A lower average $n_{H_2O}$ in one eukaryotic organism, humans, is apparent in comparison to *E. coli* (Sect. 3.3) and in the positive values of $n_{H_2O}$ for most of the metagenomic and metatranscriptomic datasets considered here (see Figs. 2–4) (recall that the mean for human proteins was defined to be zero). These preliminary observations suggest that the evolution of multicellularity may be accompanied by
an overall decrease in stoichiometric hydration state.

Another important evolutionary transition is the emergence of heterotrophic metabolism, which is a later innovation than autotrophic core metabolism (Morowitz, 1999; Braakman and Smith, 2013). It is notable that the deeper layers of the Guerrero Negro mat show greater evidence for heterotrophic metabolism (Kunin et al., 2008); likewise, heterotrophs in the "photosynthetic fringe" in Bison Pool may outcompete the autotrophs that dominate at higher and lower temperatures (Swingley et al.,
2012). These putative heterotroph-rich zones show locally lower values of $n_{H_2O}$ (Fig. 2a). If decreasing stoichiometric hydration state is a common theme across these major evolutionary transitions, then the relatively high $n_{H_2O}$ in the proteomes of organisms carrying the ancestral nitrogenase Nif-D (Fig. 2a) is not unexpected.

### 4.3   Compositional trends in rivers, lakes, and hypersaline environments

The Amazon river and ocean plume provide another example of a freshwater to marine transition, with salinities that range
from below the scale of practical salinity units (PSU) in the river to 23–36 PSU in the plume (Satinsky et al., 2014, 2015). We used published metagenomic and metatranscriptomic data for filtered samples classified as free-living (0.2 to 2.0 μm) and particle-associated (2.0 to 156 μm) (Satinsky et al., 2014, 2015). River samples form a tight cluster on a plot of stoichiometric hydration state against carbon oxidation state of proteins, and the free-living size fraction of plume samples is scattered over lower $Z_C$ whereas the particle-associated fraction shows very low values of $n_{H_2O}$ (Fig. 4a). For metatranscriptomes, there is a
noticeable decrease of $n_{H_2O}$ but little difference in carbon oxidation state (Fig. 4b), and the particle-associated samples again exhibit a generally lower $n_{H_2O}$ than the free-living samples.

To continue the investigation, we also considered data used in a previous comparative study and data for hypersaline environments including evaporation ponds (salterns) and lakes in desert areas. Eiler et al. (2014) characterized microbial communities using metagenomic data for various freshwater samples (lakes in the USA and Sweden) and marine locations. For hypersaline
settings, we used metagenomic data from the Santa Pola salterns in Spain (Ghai et al., 2011; Fernandez et al., 2013), natural soda lakes of the Kulunda Steppe in Serbia (Vavourakis et al., 2016), and South Bay salterns in California, USA (Kimbrel et al., 2018). The compositional analysis reveals a relatively low $n_{H_2O}$ of proteins inferred from the marine metagenomes compared to freshwater samples in the Eiler et al. dataset (Fig. 4c). Surprisingly, hypersaline metagenomes have ranges of $n_{H_2O}$ of proteins that are similar to marine environments, but considerably higher $Z_C$ (Fig. 4c). To interpret these results, we considered other
factors that are known to influence the amino acid compositions of proteins in halophiles.

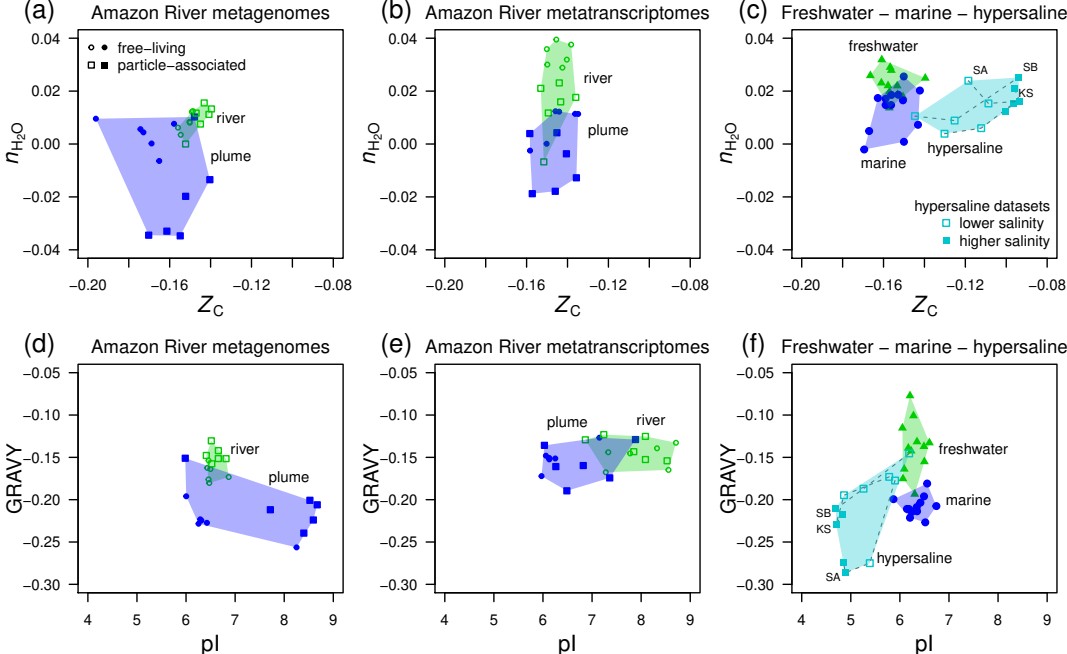

**Figure 4.** Compositional analysis and hydropathicity and isoelectric point calculations for proteins from the Amazon River and plume and other metagenomes. Samples representing freshwater, marine, and hypersaline environments are indicated by the colored convex hulls. **(a)** Metagenomic and **(b)** metatranscriptomic data for particle-associated and free-living fractions from the lower Amazon River (Satinsky et al., 2015) and plume in the Atlantic Ocean (Satinsky et al., 2014). **(c)** Freshwater (lakes in Sweden and USA) and marine metagenomes considered in a previous comparative study (Eiler et al., 2014) and metagenomes from hypersaline environments including Kulunda Steppe soda lakes in Siberia, Russia (Vavourakis et al., 2016) (KS), Santa Pola salterns in Spain (Ghai et al., 2011; Fernandez et al., 2013) (SA), and salterns in the South Bay of San Francisco, CA, USA (Kimbrel et al., 2018) (SB). Plots **(d-f)** show values of average hydropathicity (GRAVY) and isoelectric point (pI) of proteins for the same datasets.

"Salt-in" halophilic organisms have proteins with relatively low isoelectric point that remain soluble at high salt concentrations (Ghai et al., 2011). Notably, proteins with a lower pI also tend to have relatively high $Z_C$ due to higher abundances of aspartic acid and glutamic acid, which are relatively oxidized (see Amend and Shock, 1998, Dick, 2014, and Fig. 1). Consequently, the lower pI characteristic of "salt-in" organisms is also associated with an increase of carbon oxidation state. Because

of the large pI differences (Fig. 4f), the increase of $Z_C$ in hypersaline environments can not be interpreted as an indicator of an environmental redox gradient.

Some halophilic organisms are also noted to have proteins that are less hydrophobic, with lower values of GRAVY (Paul et al., 2008; Boyd et al., 2014). Because hydrophobic amino acids have relatively low values of $Z_C$ (Dick, 2014), GRAVY and $Z_C$ for proteins are negatively correlated, as shown in Fig. 5a for all proteins in the *E. coli* genome. On the other hand, there

is very little correlation in these proteins between GRAVY and $n_{H_2O}$ (Fig. 5b). A small correlation between pI and $Z_C$ is also

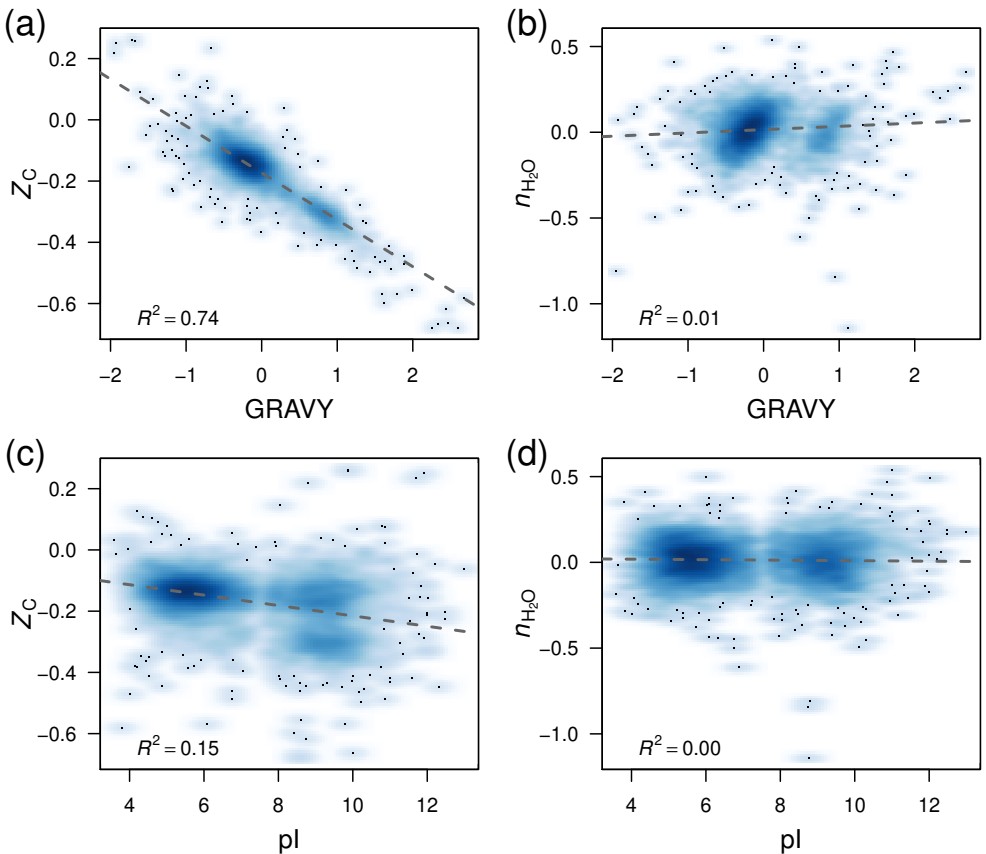

**Figure 5.** Scatterplots of $Z_C$ or $n_{H_2O}$ as a function of **(a–b)** GRAVY and **(c–d)** pI for *E. coli* proteins in the UniProt database. Linear models and $R^2$ values were calculated using the lm function in R (R Core Team, 2020).

apparent in the *E. coli* genome, in contrast to no correlation with $n_{H_2O}$ (Fig. 5c–d). Therefore, it seems likely that selection for hydrophobicity or isoelectric point are not largely responsible for trends of $n_{H_2O}$ in environmental samples.

Marine metagenomes exhibit lower hydrophobicity than most of the freshwater samples, and hypersaline metagenomes are shifted to both lower GRAVY and pI (Fig. 4f). However, there are irregular trends in the Amazon River data. Compared to

the river, the plume metagenomes exhibit lower GRAVY and either higher or lower pI (Fig. 4d). Similarly, other authors have reported that although lower pI is a signature of many hypersaline environments, it does not clearly distinguish marine from lower-salinity environments (Rhodes et al., 2010). On the other hand, the plume metatranscriptomes do show decreased pI but no major difference in GRAVY compared to river samples (Fig. 4e).

There is not enough space here to comprehensively examine all the available metagenomic data for environmental salinity

gradients. However, we have identified one dataset that gives a contrary result, and therefore offers more perspective on the compositional relationships of proteins coded by metagenomes in salinity gradients. This dataset was generated in a time-series study of microbial and viral community dynamics in a freshwater aquaculture facility ("tilapia channel" and "prebead bond")

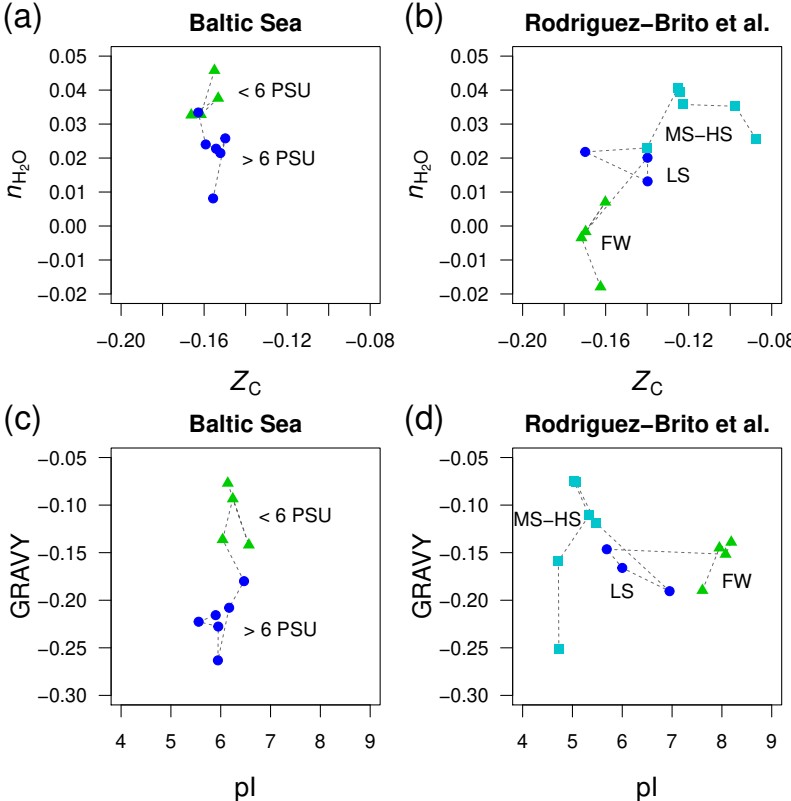

**Figure 6.** Divergent trends of $n_{H_2O}$ and $Z_C$ of proteins from metagenomes for **(a)** the Baltic Sea and **(b)** freshwater and higher-salinity samples from southern California (Rodriguez-Brito et al., 2010). The datasets from Rodriguez-Brito et al. (2010) are classified according to salinity: freshwater (FW; 3 samples at different times from the "tilapia channel" and 1 sample from the "prebead pond"), low salinity (LS; 3 samples at different times from the low salinity saltern), and hypersaline (MS–HS; 4 samples from a medium salinity and 2 from a high salinity saltern). Plots **(c)** and **(d)** show GRAVY and pI computed for the same datasets.

and low-, medium-, and high-salinity salterns in southern California (Rodriguez-Brito et al., 2010). Here, we have used only the reported microbial sequences (not the viral dataset) and considered all time points together. Contrary to our hypothesis,

the stoichiometric hydration state of proteins is lowest in the freshwater samples, which is the reverse of the trend from the Baltic Sea (Fig. 6a–b). A side-by-side comparison of the Baltic Sea and Rodriguez-Brito et al. datasets shows large changes of GRAVY in the former, but pI in the latter (Fig. 6c–d), which is another indication that these variables respond as expected only in certain ranges of salinity.

This counterexample demonstrates that the sign of differences of $n_{H_2O}$ is not predictable in all environments; however, the

large negative offset in the freshwater samples may be a signal of some other influence, perhaps related to the human control of these ponds, which are used as fish nurseries. Considering all the datasets shown in Figs. 4 and 6, there appears to be no globally consistent metric for environmental salinity gradients that can be derived from amino acid composition. If we exclude




the Rodriguez-Brito et al. (2010) dataset, then $n_{H_2O}$ exhibits a consistent decreasing trend in marine compared to freshwater samples. However, this trend does not continue into hypersaline environments.

### 4.4 Compositional analysis of differentially expressed proteins

Coming away from a picture of salinity gradients as only spatial phenomena, there is much interest in the impact of changing salinities on microbial organisms. To cite one example relevant to environmental studies, cyanobacteria respond to salt shock through stages including cell shrinkage, influx of external salts, synthesis of compatible solutes, changes in gene and protein expression, and acclimation (Qiao et al., 2013). It is also important to recognize that osmotic stress can be imposed by solutes other than NaCl; the effects of organic solutes differ in relation to their ability to permeate or depolarize cell membranes and to be sensed by cellular osmoregulatory systems (Kanesaki et al., 2002; Shabala et al., 2009; Withman et al., 2013). It is clear that microbial adaptation to changes in osmotic conditions is a dynamic process, so it is helpful to look at gene and protein expression data for a range of times and conditions that can be controlled in the lab.

We performed multiple literature searches to compile data for differential gene and protein expression in non-halophilic bacteria in NaCl or other osmotic stress conditions. As a general rule, we included only datasets with a minimum of 20 down-regulated and 20 up-regulated genes or proteins; however, smaller datasets were included if they are part of a study with larger datasets. This compilation consists of 49 transcriptomics and 29 proteomics datasets from 35 studies (note that different time points and treatments are considered as separate datasets); descriptions and references for all datasets are given in Figures S1 and S2. We assembled the lists of up- and down-regulated proteins in each dataset or, for gene expression studies, the proteins corresponding to the up- and down-regulated genes, and converted gene names or accession numbers to UniProt accessions using the UniProt mapping tool (Huang et al., 2011). The compiled data are available as CSV files in R packages (see *Code and data availability*). This is a major update to an earlier compilation of data for hyperosmotic stress experiments (Dick, 2017), but we have limited the present compilation to data for bacteria; data for osmotic stress induced by NaCl or glucose in eukaryotic cells are considered in a separate paper (Dick, 2020a).

After removing genes or proteins with unavailable or duplicated UniProt IDs and those with ambiguous differences (appearing in both the down- and up-regulated groups), the amino acid compositions computed for protein sequences downloaded from UniProt (The UniProt Consortium, 2019) were used for the compositional analysis of carbon oxidation state and stoichiometric hydration state. In Fig. 7, the values of $\Delta Z_C$ and $\Delta n_{H_2O}$ represented by empty and lettered symbols refer to median differences in individual datasets; that is, the median value for all up-regulated proteins minus the median value for all down-regulated proteins. Although there is obvious scatter in values, the $\Delta n_{H_2O}$ for proteins in transcriptomic and proteomic experiments is negative on average (Fig. 7a–b), but the differences are non-significant to marginally significant [$p = 0.215$ and $0.052$, respectively; all $p$-values were calculated for paired two-sided Student's $t$-tests using R (R Core Team, 2020)]. The compilations of gene and protein expression data also show small average $\Delta Z_C$, with $p = 0.088$ and $0.666$, respectively.

Figure 7c shows results for selected time-course experiments for osmotic stress. Note that all values are differences calculated relative to the same control (starting condition) in a given study. In transcriptomic experiments for a commensal species (*Enterococcus faecalis*), a soil bacterium (*Methylocystis* sp. strain SC2), and two pathogens (*E. coli* O157:H7 and *Salmonella*

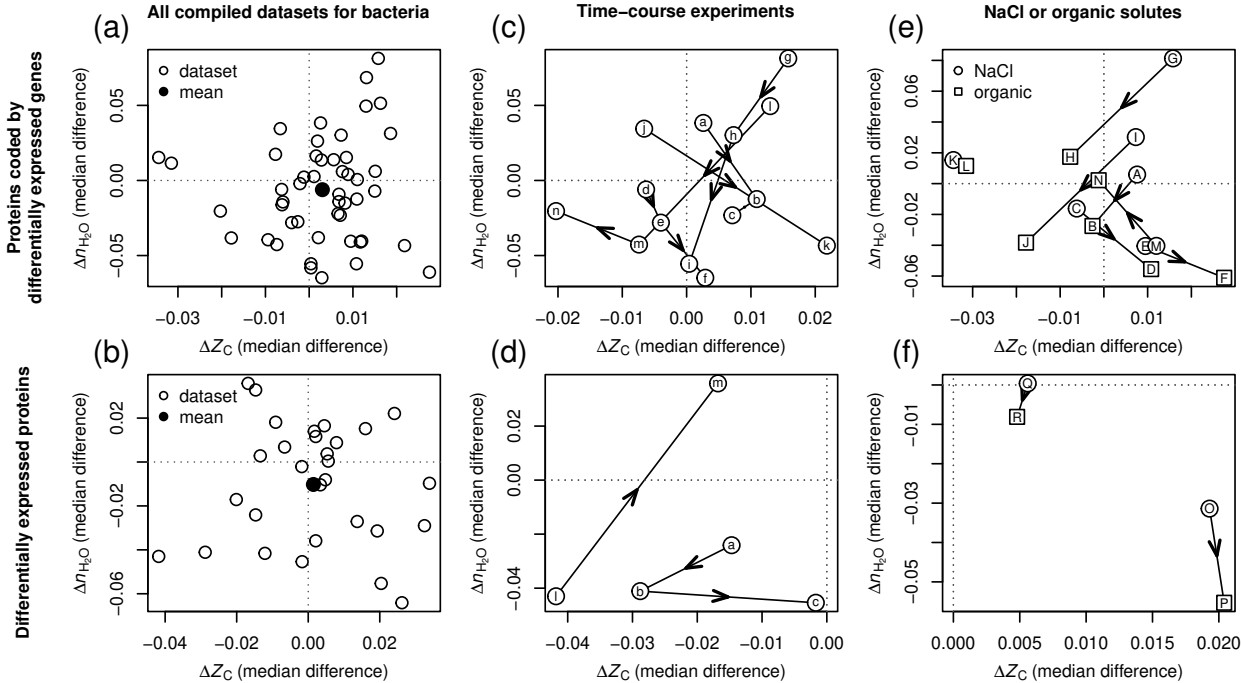

**Figure 7.** Compositional analysis of proteins in hyperosmotic stress experiments for non-halophiles. All datasets and mean value for all datasets in each compilation are shown for **(a)** proteins coded by differentially expressed genes and **(b)** differentially expressed proteins. See Figures S1 and S2 for references for all datasets. Selected time course experiments are highlighted in **(c)** and **(d)**. Points connected by lines show the progression in each experiment: a–c (30, 80, 310 min; Kocharunchitt et al., 2014), d–f (5, 30, 60 min; Solheim et al., 2014), g–i (1, 6, 24 h; Finn et al., 2015), j–k (45 min, 14 h; Han et al., 2017), l–n (24, 48, 72 h; Qiao et al., 2013) (no proteomic data available at 72 h). **(e–f)** Pairs of experiments for osmotic stress imposed by NaCl or organic solutes. The sources of data are: A–B (sorbitol; Kanesaki et al., 2002), C–D (sorbitol; Han et al., 2005), E–F (sucrose; Kohler et al., 2015), G–H (glycerol at 1 h; Finn et al., 2015), I–J (glycerol at 6 h; Finn et al., 2015), K–L (sucrose; Shabala et al., 2009), M–N (urea; Withman et al., 2013).

*enterica* serovar Typhimurium) (Solheim et al., 2014; Han et al., 2017; Kocharunchitt et al., 2014; Finn et al., 2015), there is a marked progression toward lower $n_{H_2O}$ of the associated proteins with time. In a transcriptomic experiment for salt stress in *Synechocystis* sp. PCC 6803 (Qiao et al., 2013), $\Delta n_{H_2O}$ is shifted negatively between 24 and 48 h, but rises to a less negative

value at 72 h. Proteomic data are available from two of these studies, indicating that the differentially expressed proteins in *E. coli* (Kocharunchitt et al., 2014) also show decreasing $n_{H_2O}$ with time (Fig. 7d), but in the proteomic experiment for *Synechocystis* sp. PCC 6803 (Qiao et al., 2013), $\Delta n_{H_2O}$ changes sign from negative to positive between 24 and 48 h.

Perhaps the most striking result to emerge from this analysis is the strong dehydrating signal associated with osmotic stress imposed by organic solutes. We compared pairs of datasets from the same study for NaCl and another solute at concentrations

that give similar total osmolalities. Transcriptomic data for sorbitol (Kanesaki et al., 2002; Han et al., 2005), sucrose (Kohler





**Table 2.** Halophilic organisms, growth conditions, number of differentially expressed proteins, and sources of data for hypoosmotic and hyperosmotic stress experiments. Units for NaCl concentrations are taken from the references; approximate conversions between molarity and weight percent are 1 M NaCl ≈ 6%, 2.5 M NaCl ≈ 13%, 4 M NaCl ≈ 20%.

| ID | Organism | Conditions |
|----|----------|------------|
| a | *Halobacterium salinarium* | 2.6M / 4.3 M NaCl |
| b | *Halobacterium salinarium* | 5.1 M / 4.3 M NaCl |
| c | *Nocardiopsis xinjiangensis* | 6% / 10% NaCl |
| d | *Nocardiopsis xinjiangensis* | 17.5% / 10% NaCl |
| e | *Tetragenococcus halophilus* | 0 M / 1 M NaCl |
| f | *Tetragenococcus halophilus* | 3.5 M / 1 M NaCl |
| g | *Haloferax volcanii* | 10.8% / 15% NaCl |
| h | *Haloferax volcanii* | 19.2% / 15% NaCl |

Data sources: (a, b) Tables 1 and 2 of Leuko et al. (2009). (c, d) Table S-1 of Zhang et al. (2016). Values of reporter intensities at each condition (6%, 10%, and 17.5% NaCl) were quantile normalized and used to compute intensity ratios (6% / 10% NaCl and 17.5% / 10% NaCl). Only proteins with expression ratios > 1.3 in either direction (Zhang et al., 2016), *p*-values < 0.05, and at least 2 peptides were included. (e, g) Tables S2 and S3 of Lin et al. (2017). (g, h) Supporting Table 1C of Jevtić et al. (2019). Only proteins with at least 2-fold expression difference and marked as significant were included.

et al., 2015), and glycerol (Finn et al., 2015) compared to controls all show a lower $\Delta n_{H_2O}$ of the associated proteins than for NaCl compared to controls (Fig. 7e). Data from the study of Finn et al. (2015) are plotted at 1 and 6 h in the experiment, indicating a time-dependent decrease as well as more negative values for glycerol than NaCl. Experiments with different strains of *E. coli* show a smaller negative difference between NaCl and sucrose (Shabala et al., 2009) and the only positive difference

for an organic solute (urea) compared to NaCl (Withman et al., 2013). The available proteomic data also show lower $n_{H_2O}$ for sucrose (Kohler et al., 2015) and glucose (Schmidt et al., 2016) compared to NaCl (Fig. 7f). Note that the latter dataset is actually a comparison between growth on glucose and glucose with NaCl; growth on glucose alone produces a lower $\Delta n_{H_2O}$ of the differentially expressed proteins. The marked decrease of $\Delta n_{H_2O}$ induced by solutes such as sorbitol, which does not permeate the plasma membrane, could follow from a higher effective osmotic pressure compared to NaCl (Kanesaki et al.,

2002). However, sucrose, which permeates but unlike NaCl does not depolarize the plasma membrane (Shabala et al., 2009), also exhibits a strong dehydrating effect.

We also considered the changes in protein expression when halophilic organisms are exposed to hyperosmotic conditions in the laboratory. Proteomic data were found for four halophilic species of bacteria and archaea for hypo- and hyperosmotic stress under changing NaCl concentrations (Leuko et al., 2009; Zhang et al., 2016; Lin et al., 2017; Jevtić et al., 2019) (Table

2). The combined data are plotted in Fig. 8a. A negative $\Delta n_{H_2O}$ of the differentially expressed proteins characterizes most of the hyperosmotic stress experiments; only *Tetragenococcus halophilus* shows a small positive value. Unexpectedly, growth

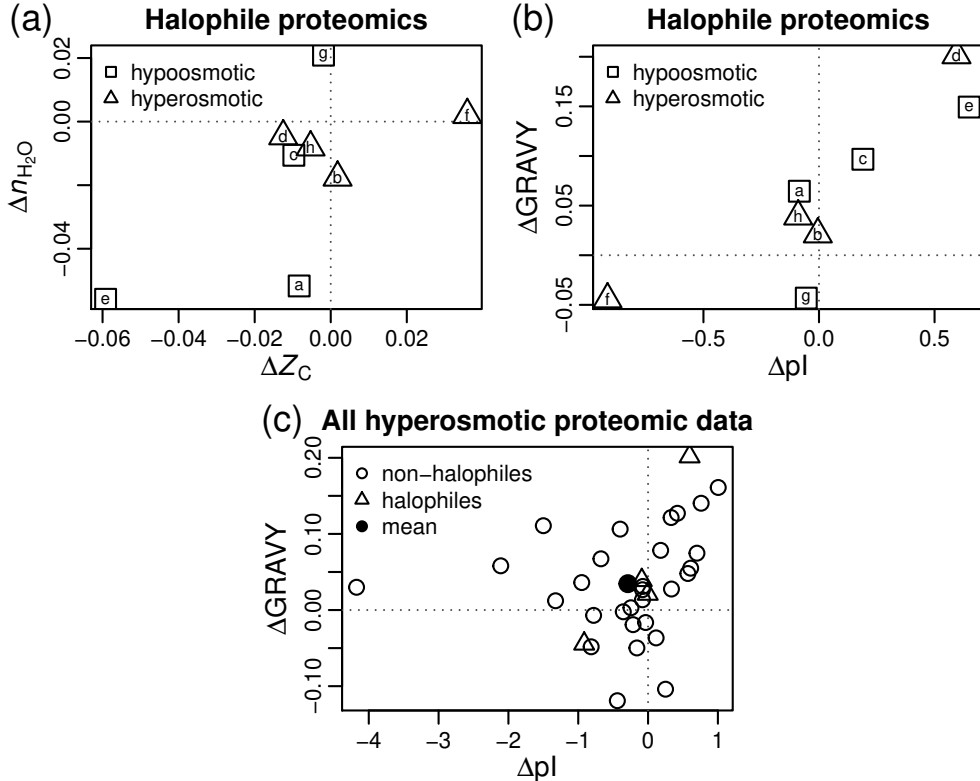

**Figure 8.** Compositional analysis of differentially expressed proteins in halophiles under hypoosmotic and hyperosmotic stress. **(a)** Median differences of $n_{H_2O}$ and $Z_C$ between up- and down-regulated proteins in hypoosmotic compared to optimal growth conditions and hyperosmotic compared to optimal growth conditions. See Table 2 for experimental conditions and references. **(b)** Median differences of GRAVY and pI for the same datasets. **(c)** Median differences of GRAVY and pI for all compiled proteomics data for hyperosmotic stress in halophiles and non-halophiles.

at NaCl concentrations below the optimal concentrations (i.e. hypoosmotic stress) in three of these organisms – the archaeon *Halobacterium salinarium* and bacteria *Nocardiopsis xinjiangensis* and *Tetragenococcus halophilus* – induces an even larger loss of $n_{H_2O}$ in the differentially expressed proteins (points labeled a, c, and e in Fig. 8a).

The median difference of GRAVY increases for differentially expressed proteins in three of the four halophilic organisms under hyperosmotic stress (Fig. 8b). Considering all the data for hyperosmotic stress in both halophiles and non-halophiles, the average value of GRAVY increases significantly (Fig. 8c; $p = 0.010$). The data also exhibit a small decrease of pI ($p = 0.100$), which is expected for halophiles, but the increase of GRAVY – that is, higher hydrophobicity – is the opposite of the evolutionary trend for proteomes of halophilic organisms (Paul et al., 2008) and the metagenomic comparisons described above.

We therefore propose that $n_{H_2O}$ is a more consistent metric, since it records decreasing hydration state with increasing salinity





in the Baltic Sea and Amazon River and plume and in differentially expressed proteins of both halophiles and non-halophiles under hyperosmotic stress.

## 5   Conclusions

Based on mass-action effects in thermodynamics, we predicted that the stoichiometric hydration state of proteins ($n_{H_2O}$) should
decrease toward higher salinity. We found that protein sequences inferred from metagenomes in regional salinity gradients, including the Baltic Sea freshwater-marine transect and Amazon River and plume, are characterized by changes of $n_{H_2O}$ in the predicted direction. However, the trend does not continue into hypersaline environments, and there are conflicting results derived from metagenomic data used in previous comparative studies: $n_{H_2O}$ decreases between freshwater lakes and marine samples (Eiler et al., 2014) but increases between freshwater aquaculture ponds and salterns (Rodriguez-Brito et al., 2010).

While biomolecular data for environmental salinity gradients reflect phylogenetic differences and evolution, laboratory experiments provide information on the physiological effects of osmotic conditions on protein expression in single organisms. Compilations of transcriptomic and proteomic data for non-halophilic organisms indicate a small decrease of $n_{H_2O}$ on average for the differentially expressed proteins in hyperosmotic stress experiments. The dehydration signal is stronger for most organic solutes (except urea) than for NaCl. Differentially expressed proteins in halophiles show a more complex response: for three
of four organisms with available data, $\Delta n_{H_2O}$ is much lower in hypoosmotic compared to hyperosmotic conditions, which is an unexpected finding.

We were also surprised to find a pattern of relatively low $n_{H_2O}$ in the interior compared to upper layers of the Guerrero Negro microbial mat and in particles compared to free-living fractions in both the Baltic Sea and Amazon River. This effect is probably associated with phylogenetic differences among the size fractions, but reduced accessibility to bulk water may be a
contributing factor. The latter possibility can be further investigated through compositional analysis of differentially expressed proteins between single-species biofilms and planktonic growth in the laboratory.

The central message of this study is that geochemical and laboratory conditions can influence, but naturally do not completely determine, the chemical compositions of proteins. The compositional analysis establishes the feasibility and the limits of using thermodynamic models to predict the biomolecular makeup of organisms in new environments. The usefulness of
multidimensional models is also apparent, since different compositional metrics, representing oxidation state and hydration state of molecules, can in some cases be associated specifically with redox and salinity gradients, respectively. The findings of this study underscore an opportunity for the integration of hydration state into evolutionary models that already consider changes in oxidation state or oxygen content of proteins (Acquisti et al., 2007; Poudel et al., 2018).

*Code and data availability.*

All metagenomic and metatranscriptomic data analyzed here were obtained from public databases using the accession numbers listed in Supplementary Table S1 for salinity gradients and Table S2 for redox gradients. The amino acid compositions of

subsampled sequences from the metagenomic and metatranscriptomic data are available in the JMDplots R package, version 1.2.2 (https://github.com/jedick/JMDplots), which is archived on Zenodo (Dick, 2020b). Specifically, the data are contained in the file `inst/extdata/gradH2O/MGP.rds`, which can be read using the R function readRDS (minimum R version: 2.3.0).

The compilation of differential gene expression data is available in the JMDplots package as xz-compressed CSV files in the directory `inst/extdata/expression/osmotic/`. The compilation of differential protein expression data is in the corresponding directory of the canprot R package, version 1.0.0 (https://cran.r-project.org/package=canprot), which is also archived on Zenodo (Dick, 2020c). The results of the compositional analysis of differential expression data, which are used for Figs. 7–8, are in the `inst/vignettes/` directories of the JMDplots and canprot packages.

The code used to make all of the figures and perform statistical testing is in the JMDplots package. The `gradH2O.Rmd` vignette in the package contains the function calls used for the figures.

*Author contributions.* JMD designed and carried out the analysis. JMD, MY and JT interpreted the results. JMD wrote the manuscript with editing input from MY and JT.

*Competing interests.* The authors declare that they have no conflict of interest.

*Acknowledgements.* We are grateful to Saroj Poudel for commenting on an earlier version of the manuscript. This work was supported by funding from the State Key Laboratory of Organic Geochemistry (Grant No. SKLOG-201928 to JD).




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
