# Peer review of "Uncovering chemical signatures of salinity gradients through compositional analysis of protein sequences"

_Biogeosciences, 2020_

## Referee Comment (RC1) · Anonymous Referee #1 · 16 Jun 2020

Dick et al. have mined the biomolecular literature to show that the composition of proteins in microorganisms reflect the salinity of their environments. In particular, their results provide evidence that the stoichiometric hydration state of amino acids is lower in many saline settings than in freshwater environments. The authors use metagenomes, metatranscriptomes and proteomes of individual organisms resulting from environmental and laboratory studies. Their method of analysis includes a rather novel technique – they assess the difference in the stoichiometric hydration state ($n\_H2O$) of theoretical formation reactions for the amino acids in different proteins (measured or inferred from metagenomes). These formation reactions are familiar to those who carry out geochemical modeling, though the choice of basis species is unusual. $H2O$ is used

as a basis species in addition to O2 and three amino acids (glutamine, glutamic acid and cysteine). To help make sense of their results, the authors also compute and compare values of the oxidation state of carbon in amino acids/proteins as well as their hydropathicities and isoelectric points. Ultimately, the authors seek to show a quantitative relationship between the composition of organisms (their biomolecules) and their environments.

I support publication of this work after some clarifying text is added in the areas noted below.

Because this work used techniques that are well known in one field (geochemical modeling) and applies them to another (biomolecular sequence analysis), it would be most helpful if the authors showed an example of the differing stoichiometric hydrations state of two proteins. Maybe this wouldn't work too well in a figure, but perhaps some combination of a table and schematic would go a long way towards explaining their methods.

The title of Table 1 should spell out what rQEC is – especially since it is conceptually and acronymically very close to QEC.

Some clarification is needed concerning the calculation of rQEC. In Table 1, the value of n_H2O for alanine is 0.369. The example for calculating n_H2O using the QEC formulation for alanine is 0.6. The correction noted in the caption for Fig. 1 to transform QEC to rQEC is 0.355. My calculator says that 0.6-0.355 = 0.245, not 0.369. Please explain.

Lines 195-196: The authors here refer to 8 amino acids by their three-letter abbreviations, but in Table 1 and in the naming of their basis species (QEC), they refer to amino acids by their one-letter abbreviations. Is there a particular reason for this difference?

It seems like the text on lines 226-227 could be better represented by an equation. This would make it easier to look back on how the stoichiometric hydration state was calculated.

Section 3.5 needs more explanation. The title of this section suggests that it's about organisms containing the Nif gene, and the authors get around to talking about these organisms, but some explanation is needed about why this gene was used as a filter for which proteomes to select (data availability?). Also, start this section with 'what' and 'why', then tell us the 'how'. It starts with 'how,' making it hard to follow.

Section 3.6 The authors should state explicitly if they did or did not take into account how temperature effects values of the isoelectric point. The same goes for using GRAVY. Amino acid pKa's and the permittivity of water certainly change with temperature.

Section 3.7 Is the sum of the 100 subsamples equivalent to $\sim$50,000 amino acids for each sample? Then what is the typical subsample density?

The beginning of Section 4.2, like in other parts of the manuscript, starts out with 'how', but should lead with what the section is all about. For instance, this paragraph should start by saying that the stoichiometric hydration state of proteins can be determined by more factors than just salinity. Instead, it starts with "Metagenomic and metatranscriptomic data for different filter size fractions are available for the Baltic Sea." This topic sentence does not reveal to the reader what this section is about and it fails to capture the point of the analyses described in the section.

Line 291 notes the "0.1–0.8 mm size fraction," but what this means isn't explained until the next section. Either explain it where it first appears or direct the reader to where it is explained. In general, the authors should be careful what they mean. When a filter fraction is noted, this could mean the DNA collected from the filtrate or that which doesn't pass through.

Perhaps an explanation for why values of n_H2O in the Rodriguez-Brito et al., 2010 data set do not follow the expected trend is that fish nurseries are extremely nutrient rich and the associated microbial communities may not be responding as they would in a typical natural system that is less persistently copiotrophic.

Many of the sentence in the Section 5 (Conclusions) should be the first sentence of the sections whose results they summarize. This would make following the text in these sections more straightforward. Tell the reader the result, then explain the supporting evidence.

Lines 371-372 – this lead sentence begins to summarize the paragraph, but then wanders away. It seems that the authors should simply note that in addition to spatial changes in salinity, there are temporal effects to changes that also merit study/consideration.

Figure 1 – what is the difference between the blue-fuzz-halo and black rectangular/square shapes in panels e, f, h and i? I'm guessing that this is due to the large number of proteins in whole proteomes, but why the difference in symbols? Same question for Fig. 5.

Figure 2. The caption says that the abbreviations and data sources for panel (a) are given in Fig 2. They are not. Panel (b) should be remade. The symbols differ in color, fill and direction, but the caption only notes what the directional difference means. Also, though I see that this plot is made at the same scale as panel (a), the result is a lot of white space and a bunch of cramped symbols connected by slightly different line styles. I've enlarged it on my external monitor and it's still hard to make sense of it.

Figure 3. It would be helpful if there was something like "–> salinity" along the x-axis.

Figure 4. Is the difference between the open and closed symbols in panels a, b, d and e that the open ones represent lower salinity samples and the closed ones higher salinity ones? If so, please state in the caption.

Figure 7. color coding time series data in panels c and e would be quite helpful

It should be noted somewhere in Table 2 that the ID and associated information are relevant to Figure 8.

The supplemental figures in S1 and S2 need captions.

---

## Referee Comment (RC2) · Anonymous Referee #2 · 5 Oct 2020

Dick et al explore (meta)genomic and (meta)transcriptomic datasets to identify a thermodynamic-based metric to characterize the effect of redox and, perhaps to a greater extent, salinity on the composition of proteins, with the argument being that minimizing energetic demands associated with protein synthesis should be a selectable phenotype. The authors compare one of several metrics, including the average oxidation state of carbon in proteins and their stoichiometric hydration state and compare these to several existing indices that have previously been shown to track with salinity gradients including the GRAVY index and the average protein isoelectric point. The results suggest that in many systems (but not all) the stoichiometric hydration state outperforms the other metrics in describing variation in proteome composition across

salinity gradients.

In general, I found the paper to be another interesting read from the primary author. However, as a microbiologist that is interested in understanding how energy availability and demand affect the distribution of microorganisms and their evolution, I would appreciate seeing a more robust effort to link the thermodynamics way of thinking (as presented here) to physiological process or mechanism that could then be used to gauge why such patterns may exist. More or less, I think this is a missed opportunity that, if executed effectively, could elevate the utility of this paper and this way of thinking. Thus, I strongly suggest the authors attempt to explain their observations at a level that makes sense to the more biologically oriented reader. As I was reading this, I could not help but think to myself how any one or several observations made sense from the level of phenotype and natural selection. The authors might consider asking themselves this same question and then speculating where possible to make this body of work a greater utility for the community.

The following list of minor comments is meant to further improve this work:

Line 1: For the average reader – what is the connection between thermodynamics and environmental variation. Lead in with this first.

Line 8: Replace "behave" with something more valid. The metric does not correlate for XXX in hypersaline environments. . .

Line 15: Communities do not adapt, populations of individuals do.

Line 26: I would not call this complementary but rather an interrelated approach since selection (imposed as an argument in previous paragraph) can and should act on the energetic demand of protein synthesis.

Line 39-40: What about the authors own work on the communities inhabiting the outflow channel at Bison Pool, Yellowstone?

Line 44-45: While I don't disagree with this assumption, at least as a first order constraint, it would be useful to relate to the reader why this assumption is made. Perhaps to avoid this confusion, the authors move this statement to below where they describe and justify their approach.

Line 58-62: This paragraph seems out of place. I suggest moving the discussion of what you did previously up in the introduction and add the last sentence of this paragraph to the end of the preceding paragraph.

Line 67: alternatives to what?

Line 305-310: I don't understand the reasoning here? Why did eukaryotes start to become important in these systems? Are there actually eukaryotes in these systems? The authors have the data to evaluate this and should evaluate it to see if the logic makes sense.

Line 315: Why would heterotroph proteomes have a lower hydration state?

Line 315-317: is there an argument to be made about why a major evolutionary transition favors a shift from higher to lower dehydration state? i.e., is this an adaptive feature that allows the latter to compete with the former from an evolutionary perspective?

Line 325: is it possible that diffusion limitation makes H2O less available to cells living nearer to a particle surface? Again, an explanation for what the observations might mean is warranted

Line 350: proteins in metagenomes

Line 360: Could this be due to aquaculture and introduction of more organic compounds/waste and its selection of heterotrophic taxa, that as stated earlier in the paper, tend to host proteomes with a lower hydration state

---

## Author Comment (AC1) · 15 Oct 2020

*Summary of Major Changes*

- Fig. 1e: Added scatter plot of $R^2$ for $n_{H_2O}$–$Z_C$ fits vs $R^2$ for $n_{O_2}$–$Z_C$ fits for all possible combinations of basis species with three amino acids, $H_2O$ and $O_2$ to illustrate the criteria for choosing basis species.

- Removed the rQEC derivation (residual-corrected values of $n_{H_2O}$); now values of $n_{H_2O}$ are taken directly from the QEC basis species (glutamine, glutamic acid, cysteine, $H_2O$, $O_2$). This change affects the scale and appearance of the plots

but does not alter the findings, except to point out that negative slopes on these plots are associated with the background correlation between $n_{H_2O}$ and $Z_C$ for amino acids.

- To visualize the background correlation between $n_{H_2O}$ and $Z_C$, guidelines parallel to the fit for amino acids have been added to the plots in Figs. 3, 5, and 6.

- Added Figure 2 with schematic of $Z_C$ and $n_{H_2O}$ calculations.

- Redrew Fig. 7 to plot (a) time or (b) type of solute on horizontal axis.

*Point-by-point Response to Anonymous Referee #1*

**Dick et al. have mined the biomolecular literature to show that the composition of proteins in microorganisms reflect the salinity of their environments. In particular, their results provide evidence that the stoichiometric hydration state of amino acids is lower in many saline settings than in freshwater environments. The authors use metagenomes, metatranscriptomes and proteomes of individual organisms resulting from environmental and laboratory studies. Their method of analysis includes a rather novel technique – they assess the difference in the stoichiometric hydration state (n_H2O) of theoretical formation reactions for the amino acids in different proteins (measured or inferred from metagenomes). These formation reactions are familiar to those who carry out geochemical modeling, though the choice of basis species is unusual. These formation reactions are familiar to those who carry out geochemical modeling, though the choice of basis species is unusual. H2O is used as a basis species in addition to O2 and three amino acids (glutamine, glutamic acid and cysteine).**

The manuscript has been revised to show the reasons for this choice of basis species more clearly; in particular, Figure 1 now includes a plot comparing all possible choices of basis species that were considered within our constraints.

**To help make sense of their results, the authors also compute and compare values of the oxidation state of carbon in amino acids/proteins as well as their hydropathicities and isoelectric points. Ultimately, the authors seek to show a quantitative relationship between the composition of organisms (their biomolecules) and their environments.**

Thank you for the thorough review and constructive suggestions. We respond to each point below.

**Because this work used techniques that are well known in one field (geochemical modeling) and applies them to another (biomolecular sequence analysis), it would be most helpful if the authors showed an example of the differing stoichiometric hydrations state of two proteins. Maybe this wouldn't work too well in a figure, but perhaps some combination of a table and schematic would go a long way towards explaining their methods.**

Added Figure 2: Schematic of $n_{H_2O}$ and $Z_C$ calculations for one protein. The selected protein is chicken lysozyme (UniProt ID: LYSC_CHICK), which should be familiar to most protein chemists as it is historically one of the most extensively characterized proteins in the laboratory. The schematic represents the amino acid composition, chemical formula, and numerical results for this protein. It should be clear that the specific result depends on the amino acid composition of the protein, so we have included only one protein for clarity..

**The title of Table 1 should spell out what rQEC is – especially since it is conceptually and acronymically very close to QEC.**

The rQEC derivation was so named because it involved "residual-corrected" values of $n_{H_2O}$ obtained from the QEC basis species (glutamine, glutamic acid, cysteine, $H_2O$, $O_2$). We have removed the rQEC derivation from the revised manuscript and instead just use the coefficients from the QEC basis species without modification (see below).

**Some clarification is needed concerning the calculation of rQEC. In Table 1, the value of n_H2O for alanine is 0.369. The example for calculating n_H2O using the QEC formulation for alanine is 0.6. The correction noted in the caption for Fig. 1 to transform QEC to rQEC is 0.355. My calculator says that 0.6-0.355 = 0.245, not 0.369. Please explain.**

The rQEC derivation was made in two steps: (1) computing the residuals of the linear fits between $n_{H_2O}$ (from the QEC basis species) and $Z_C$; (2) subtracting a constant from the residuals. Step 1 can be thought of as a baseline or residual correction and Step 2 as a recentering operation. Therefore, the calculation for alanine is not 0.6 – 0.355, but rather [the residual between the fitted line and 0.6] – 0.355.

The criteria we consider in choosing the basis species are that (1) $n_{H_2O}$ of amino acids should have very little correlation with $Z_C$, (2) $n_{O_2}$ of amino acids should be strongly correlated with $Z_C$, and (3) the basis species should represent metabolites with high network connectivity.

The derivation of rQEC was meant to "fine-tune" the QEC basis species in order to satisfy criterion (1) above, but we realize in retrospect that this derivation is not theoretically justified, since rQEC loses the important quality that $n_{H_2O}$ should directly quantify the stoichiometry of thermodynamic components (basis species) in overall chemical reactions.

We have added a new panel to Figure 1 that shows the $R^2$ values for $n_{H_2O}$–$Z_C$ and $n_{O_2}$–$Z_C$ fits for all possible combinations of three amino acids with $H_2O$ and $O_2$. QEC is in the lower right corner of this plot and is nearly optimal. Although some other sets of basis species have even lower $R^2$ values for $n_{H_2O}$–$Z_C$ fits, and slightly higher $R^2$ values for $n_{O_2}$–$Z_C$ fits, they consist of amino acids (e.g. tryptophan and tyrosine) that are not central metabolites. On the other hand, glutamine and glutamic acid are more desirable because of their major roles in metabolism (criterion #3 above). Therefore, QEC appears to be the most reasonable choice of all the basis species we considered

here.

We note, however, that QEC still carries a small negative correlation between $n_{H_2O}$ and $Z_C$ for amino acids. In the revised manuscript, we do not attempt to remove this background correlation, as was done previously with rQEC. Instead, we revised the description of Fig. 3 [emphasis indicates added text]:

The trends *of carbon oxidation state* described above are visible in the scatter plot in Fig. 3*, with an added dimension: stoichiometric hydration state. The guidelines in this plot are parallel to the $n_{H_2O}$–$Z_C$ trend for amino acids (Fig. 2); their slope represents the background correlation between $n_{H_2O}$ and $Z_C$ that is inherent in the stoichiometric analysis. Sample data for Bison Pool and the submarine vents are distributed parallel to these guidelines. Therefore, the decrease of $n_{H_2O}$ along these redox gradients can be attributed to the background correlation in the stoichiometric analysis, and the differences between samples within each dataset are specifically associated with changes in carbon oxidation state and not stoichiometric hydration state*. This is an expected outcome, as the redox gradients considered here do not have large changes in salinity. . . .

**Lines 195-196: The authors here refer to 8 amino acids by their three-letter abbreviations, but in Table 1 and in the naming of their basis species (QEC), they refer to amino acids by their one-letter abbreviations. Is there a particular reason for this difference?**

The three-letter abbreviations seem more fitting for a sentence structure, but the one-letter abbreviations save space in the table and are more appropriate for forming acronyms. For consistency we have changed this sentence to use the one-letter abbreviations.

**It seems like the text on lines 226-227 could be better represented by an equation. This would make it easier to look back on how the stoichiometric hydration state was calculated.**

The equations for computing $n_{H_2O}$ and $Z_C$ from amino acid composition have been added here.

**Section 3.5 needs more explanation. The title of this section suggests that it's about organisms containing the Nif gene, and the authors get around to talking about these organisms, but some explanation is needed about why this gene was used as a filter for which proteomes to select (data availability?). Also, start this section with 'what' and 'why', then tell us the 'how'. It starts with 'how,' making it hard to follow.**

Added at the beginning of this paragraph: "[*what*] In a separate study, Poudel et al. (2018) used carbon oxidation state as a metric for comparing proteomes of organisms containing the nitrogenase gene (Nif). [*why*] The evolution of these organisms is associated with rising atmospheric oxygen through geological history. In order to replicate their results, ..." [*how*: rest of the paragraph]

**Section 3.6 The authors should state explicitly if they did or did not take into account how temperature effects values of the isoelectric point. The same goes for using GRAVY. Amino acid pKa's and the permittivity of water certainly change with temperature.**

Added: "The pK values used for calculating pI (Bjellqvist et al., 1993, 1994) and transfer free energies used in the derivation of the GRAVY scale (Kyte and Doolittle, 1982) correspond to 25 °C and 1 bar and no attempt was made here to account for the temperature effects on these properties."

**Section 3.7 Is the sum of the 100 subsamples equivalent to ∼50,000 amino acids for each sample? Then what is the typical subsample density?**

No, each subsample (not the sum of them) has ca. 50,000 amino acids. Reworded this as: "The number of sequences included in each subsample was chosen to give a total length closest to 50,000 amino acids on average." Also added these lines: "The

subsample density, or number of sequences included in each sample, depends on the average length of the metagenomic or metatranscriptomic sequences and is listed in Tables S1 and S2. This number ranges from 251 for the dataset with the highest mean protein fragment length (199.1; metagenome of hot-spring source of Bison Pool) to 1696 for the dataset with the lowest mean protein fragment length (29.5; metatranscriptome of site GS684 in the Baltic Sea)."

**The beginning of Section 4.2, like in other parts of the manuscript, starts out with 'how', but should lead with what the section is all about. For instance, this paragraph should start by saying that the stoichiometric hydration state of proteins can be determined by more factors than just salinity. Instead, it starts with "Metagenomic and metatranscriptomic data for different filter size fractions are available for the Baltic Sea." This topic sentence does not reveal to the reader what this section is about and it fails to capture the point of the analyses described in the section.**

Inserted a new "topic paragraph" for this section including the recommended topic sentence [emphasized text moved from Conclusion as also recommended]: "The stoichiometric hydration state of proteins can be influenced by factors other than just salinity. Previous authors have observed large differences between free-living and particle-associated microbial communities, which may be due in part to anoxic conditions arising from limited diffusion in particles (Simon et al., 2014). *As described below, we found a trend of relatively low $n_{H_2O}$ in particles compared to free-living fractions in both the Baltic Sea and Amazon River. This effect is probably associated with phylogenetic differences among the size fractions, but reduced accessibility to bulk water may be a contributing factor. Further support for the possible influence of physical accessibility is reduced $n_{H_2O}$ in the interior compared to upper layers of the Guerrero Negro microbial mat.*"

**Line 291 notes the "0.1–0.8 mm size fraction," but what this means isn't explained until the next section. Either explain it where it first appears or direct the reader**

**to where it is explained. In general, the authors should be careful what they mean. When a filter fraction is noted, this could mean the DNA collected from the filtrate or that which doesn't pass through.**

Added emphasized text: "*For the Baltic Sea metagenomes and metatranscriptomes,* the 0.1–0.8 mm and 0.8–3.0 mm size fractions *of particles that don't pass through the filter, which are used for subsequent DNA extraction and sequencing,* represent free living bacteria, while the 3.0–200 mm fraction contains particle-associated bacteria with average larger genome sizes and greater inferred metabolic and regulatory capacity (Dupont et al., 2014)."

**Perhaps an explanation for why values of n_H2O in the Rodriguez-Brito et al., 2010 data set do not follow the expected trend is that fish nurseries are extremely nutrient rich and the associated microbial communities may not be responding as they would in a typical natural system that is less persistently copiotrophic.**

Added: "Specifically, the microbial communities in the aquaculture ponds may not be responding as they would in a typical natural system that is less nutrient-rich."

Also added this text after the analysis of the differentially expressed proteins in laboratory experiments: "The large negative shift of $\Delta n_{H_2O}$ associated with most organic solutes compared to NaCl lends support to the notion that high organic loading could contribute to the relatively low $n_{H_2O}$ of protein sequences from metagenomes of freshwater aquaculture ponds (Fig. 6b)."

See also the related response to Referee #2; the suggestion was made that the lower $n_{H_2O}$ could be associated with a greater abundance of heterotrophs (due to input of organic compounds), as noted previously in this paper for heterotroph-rich zones in other systems (Bison Pool, Guerrero Negro microbial mat).

**Many of the sentence in the Section 5 (Conclusions) should be the first sentence of the sections whose results they summarize. This would make following the**

**text in these sections more straightforward. Tell the reader the result, then explain the supporting evidence.**

We have applied this recommendation by moving the summary about particle size to the beginning of the "Multifactorial hydration effects" section (see above) and the summary about laboratory experiments to the "Compositional analysis of differentially expressed proteins" section (see below). The remainder of the Conclusion has been revised to give a concise summary and synthesis.

**Lines 371-372 – this lead sentence begins to summarize the paragraph, but then wanders away. It seems that the authors should simply note that in addition to spatial changes in salinity, there are temporal effects to changes that also merit study/consideration.**

We have replaced the first two sentences of this paragraph with the topic sentence taken from the Conclusion: "While biomolecular data for environmental salinity gradients reflect phylogenetic differences and evolution, laboratory experiments provide information on the physiological effects of osmotic conditions on protein expression in particular organisms." Note that this lead paragraph also alludes to temporal effects ("dynamic process"), but the section also includes data on different solutes and other experiments not specifically dealing with time-course changes, so the whole section is introduced with "physiological effects of osmotic conditions on protein expression in particular organisms".

**Figure 1 – what is the difference between the blue-fuzz-halo and black rectangular/square shapes in panels e, f, h and i? I'm guessing that this is due to the large number of proteins in whole proteomes, but why the difference in symbols? Same question for Fig. 5.**

According to the documentation for the "smoothScatter" function in R, the blue colors are a "smoothed color density representation of a scatterplot" and the black symbols are points in the low-density region, which can be used to identify outliers. These

plots have been removed from Fig. 1 in the revision; likewise, the former Fig. 5 has been removed because it did not add much to the paper. (These scatter plots showed whole-proteome data for human and *E. coli*, which are not directly relevant to the environmental salinity gradients considered here.)

**Figure 2. The caption says that the abbreviations and data sources for panel (a) are given in Fig 2. They are not.**

Thanks for pointing this out; the abbreviations and data sources are now given here. In addition, an outline has been added to the point for proteomes from Nif-A organisms to indicate that they tend to occupy more oxidized environments compared to the other nitrogenase-bearing organisms (Poudel et al., 2018).

**Panel (b) should be remade. The symbols differ in color, fill and direction, but the caption only notes what the directional difference means. Also, though I see that this plot is made at the same scale as panel (a), the result is a lot of white space and a bunch of cramped symbols connected by slightly different line styles. I've enlarged it on my external monitor and it's still hard to make sense of it.**

Panel (b) has been made less crowded by splitting the data into two panels (surface samples: panel b; deeper samples: panel c) and the scale was adjusted to remove white space.

**Figure 3. It would be helpful if there was something like "–> salinity" along the x-axis.**

Added "→ higher salinity →" to the axis label.

**Figure 4. Is the difference between the open and closed symbols in panels a, b, d and e that the open ones represent lower salinity samples and the closed ones higher salinity ones? If so, please state in the caption.**

Yes, the open symbols represent river samples (lower salinity) and the closed ones represent plume samples (higher salinity). The words "river" and "plume" have been

added to the legend to make this clear.

**Figure 7. color coding time series data in panels c and e would be quite helpful. It should be noted somewhere in Table 2 that the ID and associated information are relevant to Figure 8.**

The figure has been redrawn so that log(time, minutes) is now on the horizontal axis. This makes the multiple time series experiments easy to distinguish from each other. Color and symbol shape are used here to represent the proteomics experiments.

Table 2 and former Fig. 8 for halophiles have been removed. Now the data for protein expression in halophiles under hyperosmotic stress are highlighted in Fig. 7 (red triangles) and are referenced in Fig. S3.

**The supplemental figures in S1 and S2 need captions.**

Added captions:

Figure S1: Transcriptomics data for non-halophilic bacteria in hyperosmotic stress experiments. The plots show median differences of compositional metrics, GRAVY, and pI for proteins coded by the differentially expressed genes, [. . . ]

Figure S2: Proteomics data for non-halophilic bacteria in hyperosmotic stress experiments. The plots show median differences of compositional metrics, GRAVY, and pI for the differentially expressed proteins, [. . . ]

Figure S3: Proteomics data for halophilic archaea in osmotic stress experiments. For completeness, data for both hyperosmotic (circles) and hypoosmotic (squares) experiments, which are reported together in the proteomics studies, are shown here, but only hyperosmotic stress data are used in the manuscript. The plots show median differences of compositional metrics, GRAVY, and pI for the differentially expressed proteins, [. . . ]

[. . . all captions . . . ] i.e. median value for all up-regulated proteins minus median value

for all down-regulated proteins in each dataset. Data sources, indicated by letters, are described in the following table and footnotes. Reference keys in the table, derived from the first letters of the authors' surnames and publication year, correspond to file names used for the datasets in the canprot package.

*Other Changes*

- Proteomes of Nif-bearing organisms are now made using RefSeq release 201 of July 2020, updated from release 95 of July 2019. The update decreases the number of matching organisms slightly (Nif-A: down 2 to 155; Nif-B: down 1 to 68), but does not noticeably alter the calculated $Z_C$ and $n_{H_2O}$ shown in Fig. 3.

- List specific proteins used for comparison of GRAVY and pI calculations with ProtParam (UniProt IDs: LYSC_CHICK, RNAS1_BOVIN, AMYA_PYRFU).

- Removed human and *E. coli* proteome plots (panels formerly in Fig. 2 and former Fig. 5).

- An additional bacterial proteomics dataset for hyperosmotic stress was included (Huang et al., 2018 referenced in Figure S2).

- Removed table (former Table 2) and plots (former Fig. 8) for halophile protein expression datasets. The halophile proteomics data for hyperosmotic stress are now shown in Fig. 7, and Figure S3 has been added to give references for the data. Hypoosmotic stress experiments are no longer analyzed in the manuscript, but are included in Figure S3 for completeness.

- Added reference that urea permeates cells and is not hypertonic (Burg et al., 2007).
* * *

---

## Author Comment (AC2) · 15 Oct 2020

*Point-by-point Response to Anonymous Referee #2*

**In general, I found the paper to be another interesting read from the primary author. However, as a microbiologist that is interested in understanding how energy availability and demand affect the distribution of microorganisms and their evolution, I would appreciate seeing a more robust effort to link the thermodynamics way of thinking (as presented here) to physiological process or mechanism that could then be used to gauge why such patterns may exist. More or**

[Figure]

**less, I think this is a missed opportunity that, if executed effectively, could elevate the utility of this paper and this way of thinking. Thus, I strongly suggest the authors attempt to explain their observations at a level that makes sense to the more biologically oriented reader. As I was reading this, I could not help but think to myself how any one or several observations made sense from the level of phenotype and natural selection. The authors might consider asking themselves this same question and then speculating where possible to make this body of work a greater utility for the community.**

Thank you for your detailed attention to the concepts and analysis in our paper and your suggestions for improving the work. We respond to the main critiques below:

**1) a more robust effort to link the thermodynamics way of thinking (as presented here) to physiological process or mechanism**

This is an ongoing challenge. An obstacle (which could also be seen as a "missed opportunity") is that the thermodynamic way of thinking deals with energetic differences between two states of a system; without further (i.e. extra-thermodynamic) constraints, it is not possible to explicitly deal with underlying mechanisms in a thermodynamic model. This paper does not attempt to build such a thermodynamic model, but uses thermodynamics as a guiding concept. A major application of thermodynamics in geochemistry is to describe and predict compositional changes in a system, e.g. the distribution of aqueous species and mineral phases with different chemical formulas. The aim of this paper is to develop a framework for describing compositional changes in geobiochemical systems, and one of the first challenges is to recognize that the most appropriate descriptive variables are probably different from inorganic geochemical systems. We present our conceptual arguments that oxidation and hydration state should be considered as primary variables, develop metrics that quantify them, and use the metagenomic data to explore how these metrics respond to environmental gradients of salinity and redox conditions. Clearly, this is far from the sophisticated applications of thermodynamics in geochemistry, but it serves as a step toward

a broader appreciation that compositional changes are not random, but are aligned with environmental conditions. That should motivate the development of more rigorous thermodynamic models in future studies.

As a partial response to the request for a more mechanistic understanding, it can be noted that Fig. 7 has been redrawn to place time on the horizontal axis. With this change, it should be more apparent that the chemical composition of the differentially expressed proteins changes dynamically in laboratory experiments.

**2) I strongly suggest the authors attempt to explain their observations at a level that makes sense to the more biologically oriented reader.**

The paper uses some technical language from physical chemistry and thermodynamics by necessity, and these technical terms are defined when introduced. These concepts are used to quantitatively analyze metagenomic datasets that are chosen to represent well-known regional gradients. The analysis of laboratory data includes protein expression in response to salt and osmotic shock. Therefore, the core of the paper is concerned with biological phenomena in an environmental context. The mixing of biological data and physicochemical metrics is what makes this paper unique; removing the quantitative language would eliminate its main contribution.

We note that the entire section on "Conceptual background" was added in a previous revision (before submission to this journal) to make the paper more accessible to biologists. The paragraphs here deal with issues about intracellular conditions, amino acid composition, distinction with polymerization reactions, selection for structural stability of proteins, other variables like temperature and pH, and relation of the basis species to biosynthetic mechanisms. However, our intention is not to write a theoretical paper but rather to present a coherent set of data analyses to convince the reader that compositional differences of proteins have a basic significance in geobiochemical systems.

**3) I could not help but think to myself how any one or several observations made sense from the level of phenotype and natural selection. The authors might con-**

**sider asking themselves this same question and then speculating where possible to make this body of work a greater utility for the community.**

We believe that the analysis of laboratory experiments of protein expression in salt and osmotic conditions does provide basic information about the effects of the environment on the observable characteristics of cells. Admittedly, this is only one aspect of the phenotype, and other types of experiments could be considered, like gene expression, metabolomes, and metabolic fluxes, but analysis of those types of data is out of the scope of this paper.

A relevant finding from a paper in preparation is that the stoichiometric hydration state of differentially expressed proteins is strongly decreased in 3D (tissue-like) compared to 2D (monolayer) culture conditions of eukaryotic cells (Dick, 2020). The lower $n_{H_2O}$ in 3D culture has some similarity to the observation in this study that metagenome-inferred proteins in particles tend to have lower hydration state compared to free-living fractions. These responses could plausibly be associated with lower water accessibility in the interiors of particles in environmental samples and in spheroids in 3D cell culture.

Regarding the evolutionary implications, another paper is in early preparation that shows the hydration and oxidation state computed for whole proteomes of phylogenetic groups predicted from the RefSeq database. This tree-like view of the chemical composition no doubt would help solidify the relevance of the physicochemical concepts used here to biological systems. That is being developed for a separate paper with its own set of data analysis of microbial community composition and it is too early to cite the results in this paper.

**The following list of minor comments is meant to further improve this work: Line 1: For the average reader – what is the connection between thermodynamics and environmental variation. Lead in with this first.**

Added text in Abstract: "Prediction of the direction of change of a system under specified environmental conditions is one reason for the widespread utility of thermodynamic

models in geochemistry."

**Line 8: Replace "behave" with something more valid. The metric does not correlate for XXX in hypersaline environments. . .**

Changed "behave" to "respond".

**Line 15: Communities do not adapt, populations of individuals do.**

Changed "communities" to "populations".

**Line 26: I would not call this complementary but rather an interrelated approach since selection (imposed as an argument in previous paragraph) can and should act on the energetic demand of protein synthesis.**

Changed "complementary" to "interrelated".

**Line 39-40: What about the authors own work on the communities inhabiting the out flow channel at Bison Pool, Yellowstone?**

Added references and reworded the sentence for better context: "The oxidation state of proteins as well as lipids has been shown to be associated with oxidation-reduction (redox) gradients in a hot spring (Dick and Shock, 2011; Boyer et al., 2020), but so far energetic models have not been broadly adopted as a tool for relating metagenomic and geochemical data."

**Line 44-45: While I don't disagree with this assumption, at least as a first order constraint, it would be useful to relate to the reader why this assumption is made. Perhaps to avoid this confusion, the authors move this statement to below where they describe and justify their approach.**

This sentence has been moved down to the second point in the "Conceptual background" section, following the reference about missing hydrogen and oxidation state in stoichiometric models (Karl and Grabowski, 2017).

**Line 58-62: This paragraph seems out of place. I suggest moving the discussion of what you did previously up in the introduction and add the last sentence of this paragraph to the end of the preceding paragraph.**

The statement of previous work and what's new in this study has been moved up to the position of the former Lines 44-45 mentioned in the previous comment. The long-term research goal has been removed, because it doesn't seem to fit anywhere now.

**Line 67: alternatives to what?**

Each area of concern is summarized here as "X or Y", which seems consistent with the dictionary's definition of an alternative as "a choice between two things". To avoid confusion, this has been reworded as "six areas of concern summarized as: 1) . . . 2) . . . . . ."

**Line 305-310: I don't understand the reasoning here? Why did eukaryotes start to become important in these systems? Are there actually eukaryotes in these systems? The authors have the data to evaluate this and should evaluate it to see if the logic makes sense.**

This has been removed in the revision. The comparison of the average stoichiometric hydration state of human proteins with *E. coli* and the metagenomic data analyzed in this study provided preliminary support for the concept of a lower $n_{H_2O}$ in eukaryotes, but a more targeted data analysis is needed to strengthen this claim. Also note that the human and *E. coli* proteomes have been removed in the revised description of the choice of basis species (Fig. 1).

**Line 315: Why would heterotroph proteomes have a lower hydration state?**

There might be something basically different about their metabolic pathways in terms of water requirements at the biochemical level. Apart from *E. coli*, there probably are not many existing metabolic models that could be used to test this speculation. Added sentence: "A better understanding of these trends would require more extensive phylogenetically resolved comparisons of the compositional differences as well as biochemical (or computational) analyses of water fluxes in metabolic pathways."

**Line 315-317: is there an argument to be made about why a major evolutionary transition favors a shift from higher to lower dehydration state? i.e., is this an adaptive feature that allows the latter to compete with the former from an evolutionary perspective?**

This is certainly a valid question, but we are unable to provide a convincing mechanistic reason for why lower hydration state might offer a selective advantage. Perhaps it should be considered not as adaptation but as physical constraint, similar in a way to Gould and Lewontin (1979)'s spandrels. Structures that are physically durable, such as macromolecular complexes in organelles or larger assemblages like tissues, might be those that are relatively dry. Physical dryness (i.e. lower water content) could be a selective force for lower stoichiometric hydration state of biomolecules, but the latter by itself may have no fitness advantage.

If lower $n_{H_2O}$ turned out to characterize some evolutionary transitions, it would seem to be consistent with the postulate that "ontogeny recapitulates phylogeny" and the observation that progressive loss of water occurs in animal development through the stages of embryo, fetus, birth and growth (Moulton, 1923).

[These ideas are rather speculative, and don't specifically deal with the (non-eukaryotic) metagenomes that are analyzed here, so haven't been added to the text.]

**Line 325: is it possible that diffusion limitation makes H2O less available to cells living nearer to a particle surface? Again, an explanation for what the observations might mean is warranted.**

Particles likely provide opportunities for some amount of physical separation from the bulk aqueous phase; it's harder to pin down the molecular mechanisms. Added: "Together with the lower $n_{H_2O}$ for proteins inferred from metagenomes and metatranscriptomes in the larger size fractions from Baltic Sea samples, this could reflect a lower availability of $H_2O$ to organisms living near the particle surface due to physical separation from the bulk aqueous phase and associated diffusion limitation or lower water activity (Wang et al., 2003)."

**Line 350: proteins in metagenomes**

Changed "plume metagenomes" to "proteins in plume metagenomes".

**Line 360: Could this be due to aquaculture and introduction of more organic compounds/waste and its selection of heterotrophic taxa, that as stated earlier in the paper, tend to host proteomes with a lower hydration state**

This seems very reasonable. Added: "The microbial communities in the aquaculture ponds may not be responding as they would in a typical natural system that is less nutrient-rich. As noted above for putative heterotroph-rich zones in other systems, the lower stoichiometric hydration state could be associated with the enrichment of heterotrophic taxa, in this case due to the addition of organic compounds to the aquaculture ponds."

See also the response to Referee #1 and the revised discussion of the analysis of differentially expressed proteins: "The negative shift of $\Delta n_{H_2O}$ associated with most organic solutes compared to NaCl lends support to the notion that high organic loading could contribute to the relatively low $n_{H_2O}$ of protein sequences from metagenomes of freshwater aquaculture systems."